# Concentration dependent chromatin states induced by the bicoid morphogen gradient

**Colleen E Hannon, Shelby A Blythe, Eric F Wieschaus\***

Department of Molecular Biology, Howard Hughes Medical Institute, Princeton University, Princeton, United States

**Abstract** In *Drosophila*, graded expression of the maternal transcription factor Bicoid (Bcd) provides positional information to activate target genes at different positions along the anterior-posterior axis. We have measured the genome-wide binding profile of Bcd using ChIP-seq in embryos expressing single, uniform levels of Bcd protein, and grouped Bcd-bound targets into four classes based on occupancy at different concentrations. By measuring the biochemical affinity of target enhancers in these classes in vitro and genome-wide chromatin accessibility by ATAC-seq, we found that the occupancy of target sequences by Bcd is not primarily determined by Bcd binding sites, but by chromatin context. Bcd drives an open chromatin state at a subset of its targets. Our data support a model where Bcd influences chromatin structure to gain access to concentration-sensitive targets at high concentrations, while concentration-insensitive targets are found in more accessible chromatin and are bound at low concentrations. This may be a common property of developmental transcription factors that must gain early access to their target enhancers while the chromatin state of the genome is being remodeled during large-scale transitions in the gene regulatory landscape.

DOI: https://doi.org/10.7554/eLife.28275.001

**\*For correspondence:**
efw@princeton.edu

**Competing interests:** The authors declare that no competing interests exist.

## Introduction

During embryonic development, multicellular organisms must generate the patterned tissues of an adult organism from a single undifferentiated cell. This process requires highly regulated control of gene expression, both in developmental time and at reproducible positions in an embryo. These complex gene regulatory networks are controlled by systems of transcription factors, which bind to DNA and control the expression of genes required for development (*Levine and Davidson, 2005*). In early *Drosophila melanogaster* embryos, Bicoid (Bcd) protein forms an anterior-to-posterior gradient the embryo (*Driever and Nüsslein-Volhard, 1988b*). Bcd functions as transcriptional activator to pattern the embryo, binding to target gene enhancers and activating gene expression at distinct positions along the AP axis, corresponding to different concentrations of Bcd protein (*Driever and Nüsslein-Volhard, 1988a*; *Struhl et al., 1989*).

Recent studies of Bcd function suggest that its interaction with its targets may be more complex than the simple concentration-dependent activation originally proposed for morphogen gradients (*Wolpert, 1969*). Embryos in which the graded distribution of Bcd is disrupted still exhibit patterned expression of Bcd target genes, and these genes can be activated at lower concentrations of Bcd than these nuclei would be exposed to in a wild-type embryo (*Chen et al., 2012*; *Liu et al., 2013*; *Ochoa-Espinosa et al., 2009*). While changing Bcd dosage shifts cell fates, the shifts deviate quantitatively from those expected of strict concentration dependence, especially as expression patterns are assayed progressively later during development (*Liu et al., 2013*). Because Bcd is not the sole determinant of its target genes' expression domains, their final patterns are also influenced by other

maternal patterning systems and interactions among the Bcd targets themselves (*Chen et al., 2012*; *Jaeger, 2011*; *Löhr et al., 2009*). Chen *et al*. have shown that the posterior boundaries of Bcd target genes are positioned by a system of repressors including Runt, Krüppel, and Capicua. These studies have consequently raised doubts about the extent to which the local concentration of Bcd determines the spatial patterns of in the embryo. Further, they demonstrate that using target gene expression as a metric for Bcd function cannot directly address how information from the Bcd gradient initially establishes distinct cell fates, as these expression patterns will always be influenced by additional inputs.

Part of the difficulty in evaluating direct roles of the Bcd gradient arises from the unknown nature of the molecular mechanism by which Bcd establishes concentration thresholds at different positions along the gradient. A simple model of the positioning of Bcd target genes predicts that *cis*-regulatory elements of different genes respond to different concentrations of Bcd. Genes in the anterior would have low affinity Bcd binding sites and could therefore only be activated by high Bcd concentrations, whereas genes expressed in more posterior positions would have higher affinity binding sites (*Driever et al., 1989*). Direct measurements of Bcd binding affinity have been conducted in vitro using DNA probes (*Burz et al., 1998*; *Gao and Finkelstein, 1998*; *Yuan et al., 1996*; *Ma et al., 1996*) and have demonstrated that Bcd is able to bind cooperatively to achieve sharp concentration thresholds. While these measurements lend some support to a simple affinity model, little correlation has been shown between predicted binding site affinity and AP position of gene expression (*Ochoa-Espinosa et al., 2005*; *Segal et al., 2008*). However, neither in vitro measurements of Bcd binding nor computational predictions of binding sites can capture interactions between Bcd and its target enhancers in the context of local chromatin structure.

In this study, we sought to measure Bcd's interactions with its targets directly in the context of the early embryo. Using high throughput sequencing approaches, we measured in vivo genome-wide Bcd-DNA binding and chromatin accessibility in transgenic embryos expressing different concentrations of uniform Bcd protein. These data reveal distinct classes of DNA targets that differ in their sensitivity to Bcd concentration. We find that these classes differ both in the DNA binding motifs that they contain and in their local chromatin accessibility. We also find that Bcd influences the accessibility of a subset of its target enhancers, primarily at highly concentration-sensitive enhancers that drive gene expression in the anterior of the embryo. This leads us to a model in which target enhancers throughout the genome have a broad range of sensitivities for Bcd protein, and can therefore respond to a range of Bcd concentrations along the gradient. However, rather than arising from differences in Bcd binding site composition, these in vivo interactions are chromatin context-dependent, and Bcd influences the chromatin structure of its target enhancers.

## Results

### Bicoid target gene expression boundaries are influenced by other patterning factors, but its physical interaction with DNA targets is not

To investigate the mechanism whereby Bcd functions to pattern the AP axis, we performed chromatin immunoprecipitation followed by high throughput sequencing (ChIP-seq) to determine the genome-wide binding profile of Bcd to its targets. We performed the ChIP-seq experiments on embryos expressing GFP-tagged Bcd in a *bcd* null mutant background that were staged precisely in nuclear cycle 14 (NC14), and established a list of robust and reproducible list of 1027 peak Bcd binding regions (see *Supplementary file 1* and *Table 1*). These peaks successfully identify 63 of the 66 previously characterized Bcd target enhancers shown to drive expression patterns that span broadly across the AP axis (*Chen et al., 2012*).

As a transcriptional regulator, Bcd activates the expression of a subset of its targets whose expression domains are predominantly located in the anterior half of the embryo. In *bcd* mutant embryos, such targets are not expressed. For example, the gap genes *buttonhead* (*btd*) and *knirps* (*kni*) have anterior expression domains that are not present in *bcd* mutant embryos (*Figure 1A*). The posterior *kni* expression domain, however, is expressed in *bcd* embryos, albeit with shifted positional boundaries. These distinct domains of *kni* expression are controlled by separate enhancer elements (*Pankratz et al., 1992*; *Schroeder et al., 2004*), both of which are bound by Bcd in vivo (*Supplementary file 1*). In the absence of all maternal AP patterning inputs (*bicoid nanos hunchback*

**Table 1.** Number of Bcd ChIP-seq peaks at each step of filtering.

| Filter applied | Number of peaks | | | |
| --- | --- | --- | --- | --- |
| | Wild-Type | αTub67C > uBcd | mtrm > uBcd | bcd > uBcd |
| MACS2 | 29,090 | 15,429 | 11,812 | 38,392 |
| IDR | 9815 | 4245 | 1464 | 1329 |
| Euchromatic only | 2319 | 4123 | 1429 | 1257 |
| Common peaks (2/3) | | 4126 | | |
| ATAC-seq ratios | 2143 | 2087 | | |
| Common Peaks | 1027 | | | |

DOI: https://doi.org/10.7554/eLife.28275.002

*torsolike* [*bcd nos hb tsl*] quadruple mutants), *kni* expression is reduced to near-background levels (*Figure 1—figure supplement 1A*) and is not expressed in a posterior stripe. However, in embryos where Bcd is the sole source of maternal patterning information (*nos hb tsl* triple mutants), the *kni* posterior expression domain has a near wild-type level and anterior boundary (*Figure 1—figure supplement 1A*). The *kni* posterior domain therefore represents a second class of Bcd target gene, which depends on Bcd to determine the position of its expression but does not demonstrate an absolute requirement of Bcd for transcriptional activation.

Both classes of Bcd target genes receive positional cues both from Bcd and from other patterning systems. We considered the possibility that, given their influence on the expression domains of Bcd target genes, the posterior and terminal patterning systems may impact Bcd binding to its target enhancers in different nuclei along the AP axis. We therefore tested whether loss of the posterior and terminal systems (*nos* and *tsl*) would alter the Bcd ChIP-seq profile. We used the statistical package EdgeR (*Robinson et al., 2010*) to test for differential Bcd binding between wild-type and *nos tsl* embryos and found that we could not detect any significant change in binding at any of these 1027 regions (*Figure 1—figure supplement 1B*). Therefore, although the expression domains of Bcd target genes are ultimately influenced by inputs from other AP patterning systems, the physical interaction of Bcd with DNA occurs independently of other maternal AP patterning inputs.

## Embryos expressing bcd uniformly show developmental fates reflecting the concentration of bcd

We set out to test whether incremental changes in Bcd concentration along the gradient can be read out directly at the level of binding to target enhancers. Due to the graded distribution of Bcd, each nucleus along the AP axis is exposed to a different concentration of the protein. To measure the Bcd binding state at individual concentrations, we performed ChIP-seq on embryos expressing Bcd at single, uniform concentrations in every nucleus along the AP axis. Several previous studies have included genetic manipulations in which the Bcd gradient has been flattened to assess its activity independently of its distribution (*Chen et al., 2012*; *Driever and Nüsslein-Volhard, 1988a*; *Löhr et al., 2009*; *Ochoa-Espinosa et al., 2009*). However, genetically disrupting the gradient does not result in a total flattening, and transgenic approaches to date have not allowed for precise and reproducible control over the level of expression of the flattened Bcd. We therefore generated transgenic lines expressing GFP tagged Bcd in which the endogenous 3'UTR responsible for graded localization is flanked by FRT sites that allow it to be replaced with the unlocalized *spaghetti squash* 3'UTR. To generate different expression levels of uniform Bcd, we coupled transgenes to different maternally active promoters that yield embryos in which individual uniform Bcd concentration approximates single positions along the gradient (see *Figure 1—figure supplement 1C*).

To determine the expression levels of the uniform lines, we imaged GFP fluorescence in live embryos expressing either uniform or graded GFP-Bcd (*Gregor et al., 2007a*) (*Figure 1B*). The endogenous *bcd* promoter drives a level of uniformly expressed Bcd equivalent to that measured at approximately 65% egg length of the wild-type gradient. The *matrimony (mtrm)* and *αTubulin67C (αTub67C)* promoters drive expression levels corresponding to approximately 45% and 25% egg length, respectively. For simplicity, we refer to the uniform lines as low (*bcd* promoter), medium (*mtrm* promoter), and high (*αTub67C* promoter). (See also *Figure 1—figure supplement 1D*)

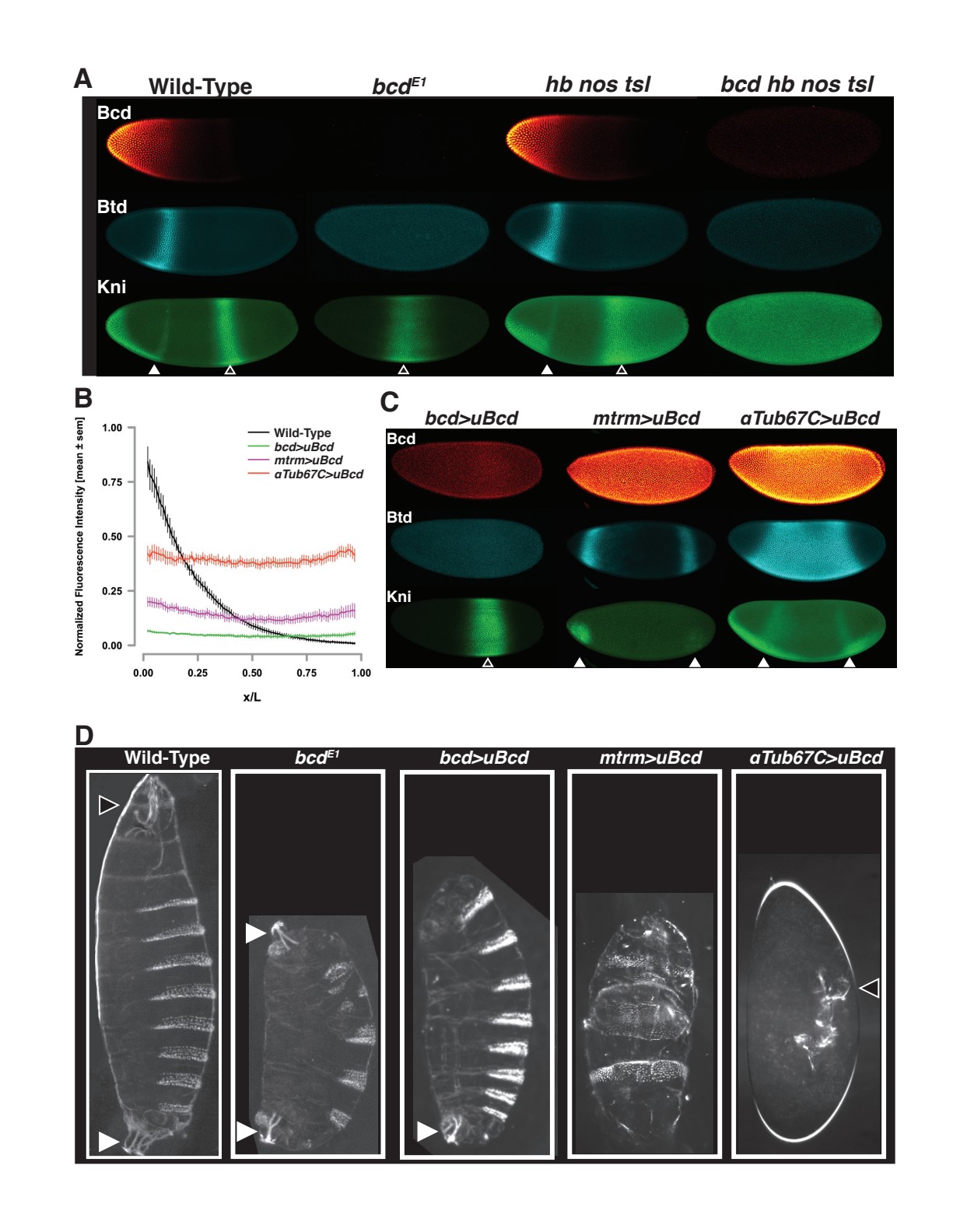

**Figure 1.** Uniform Bcd expression specifies cell fates corresponding to levels of expression. (**A**) Wild-type, *bcd* null mutant (*bcd^E1^*), and maternal *hunchback, nanos, torso-like* (*hb nos tsl*) triple mutant and *bcd hb nos tsl* mutant embryos at NC14 immunostained with antibodies against Bcd, Btd, and Kni. Embryos are oriented with anterior to the left. The anterior Kni domain (filled arrow) is absent in *bcd* but restored in *hb nos tsl* embryos, while the posterior stripe (open arrow) shifts anteriorly in in *bcd* but expands posteriorly in *hb nos tsl*. Neither Btd nor Kni exhibit patterned expression in *bcd*

*Figure 1 continued on next page*

*Figure 1 continued*

*hb nos tsl*. Images are maximum z-projections and image contrast was adjusted uniformly across the entire image for display. See *Figure 1—figure supplement 1A* for quantification of Kni intensity between genotypes. (B) Expression levels of uniform GFP-Bcd transgenic constructs relative to wild-type Bcd expression. Live embryos were imaged in during NC14, and dorsal profiles were plotted. Error bars are standard error of the mean. For wild-type, n = 23 embryos; *bcd*-uBcd n = 13; *mtrm*-uBcd n = 7; and *αTub67C*-uBcd n = 14. See also *Figure 1—figure supplement 1D* and *Table 6*. (C) Immunostaining as in (A), for each level of uniform Bcd. Anterior target gene expression is absent at the lowest level. At intermediate (*mtrm*) and high (*αTub67C*) levels of uBcd, anterior expression patterns are expanded and/or duplicated in the posterior, and posterior expression of Kni is absent. (D) Larval cuticle preparations for the indicated genotypes. Embryos are oriented with anterior at the top. Head structures are indicated with open arrows and tail structures with filled arrows. *αTub67C* >uBcd embryos develop essentially no cuticle tissue, but form only what appear to be anteriorly-derived mouth structures. *mtrm* >uBcd results in a duplication of the anterior-most abdominal denticles in the anterior and posterior of the embryo, with no clear terminal structures forming at either end. *bcd* >uBcd embryos have a normal posterior and all abdominal segments, but no thoracic or head structures. Images of individual embryos were rotated and cropped to exclude nearby embryos and air bubbles.

DOI: https://doi.org/10.7554/eLife.28275.003

The following figure supplement is available for figure 1:

**Figure supplement 1.** Target gene expression and Bcd binding in maternal matterning mutants and features the uniform Bcd transgene.

DOI: https://doi.org/10.7554/eLife.28275.004

Uniform expression of Bcd confers gene expression profiles and developmental programs representative of distinct positions along the AP axis. The head gap gene *buttonhead (btd)* is expressed in an anterior stripe in wild-type embryos (*Figure 1A*), but expands to fill the entire middle of the embryo at the highest level of uniform Bcd (*Figure 1C*). At the medium level, the Btd anterior stripe is duplicated at the posterior, and at the lowest Bcd level, it is not expressed. The gap gene *knirps*, which is expressed in an anterior domain and a posterior stripe, shows a duplication of its anterior domain in the posterior in high uniform Bcd embryos. There is also a weaker duplication at the medium Bcd level. There is no apparent anterior expression at the lowest level, but an expanded posterior stripe is present. The gene expression patterning we observe in the presence of uniform Bcd likely results from the activity of additional maternal patterning cues (*nanos* and *torso*) as well as interactions between the Bcd target genes themselves. The concentration-dependent activity of uniform Bcd is also apparent in cuticle preparations of embryos expressing the transgenic constructs. The transgenic constructs specify increasingly anterior structures along larval body plan as the concentration of Bcd increases (*Figure 1D*). These effects on the body plan indicate that the uniform Bcd transgenes can specify cell fates that reflect their relative expression levels.

## Bcd binding to genomic targets is concentration dependent

We next determined genome-wide Bcd binding profiles at each individual concentration by ChIP-seq and used these measurements to assign each of the 1027 peak regions to classes distinguished by their degree of concentration-dependent Bcd binding (*Figure 2A*). Using EdgeR (*Robinson et al., 2010*), we selected peak regions that exhibited statistically significant (FDR $\leq$ 0.05) differences in binding by performing pairwise exact tests between the three uniform Bcd concentrations. This yielded four different classes of peaks, one concentration-insensitive class, and three classes with increasing sensitivity to Bcd concentration.

The Concentration-Insensitive peak class (n = 143) shows no significant differences between any of the concentrations of uniform Bcd we tested. Concentration-Sensitive III peaks (n = 593) are significantly reduced in binding between the highest and lowest Bcd concentrations, but reductions are not significant between high and medium, or medium and low. Concentration-Sensitive II peaks (n = 138) are significantly reduced in binding at the lowest Bcd level compared to either the medium or the high levels. Finally, Concentration-Sensitive I peaks (n = 152) are significantly reduced in binding at both the medium and the low Bcd levels compared to the highest level (*Figure 2A*). These different groups suggest that Bcd binds differentially to its targets at specific concentrations, and furthermore that certain subsets of these targets are bound only in anterior nuclei whereas others are bound broadly across the entire AP axis.

Although 63 out of 66 previously characterized Bcd-dependent enhancers are identified in our ChIP peaks, the majority of the 1027 peaks identified have not been extensively examined. Within the set of known Bcd targets, there is strong correlation between position of expression and the associated Bcd sensitivity class (*Figure 2A*). To extend this observation to previously uncharacterized

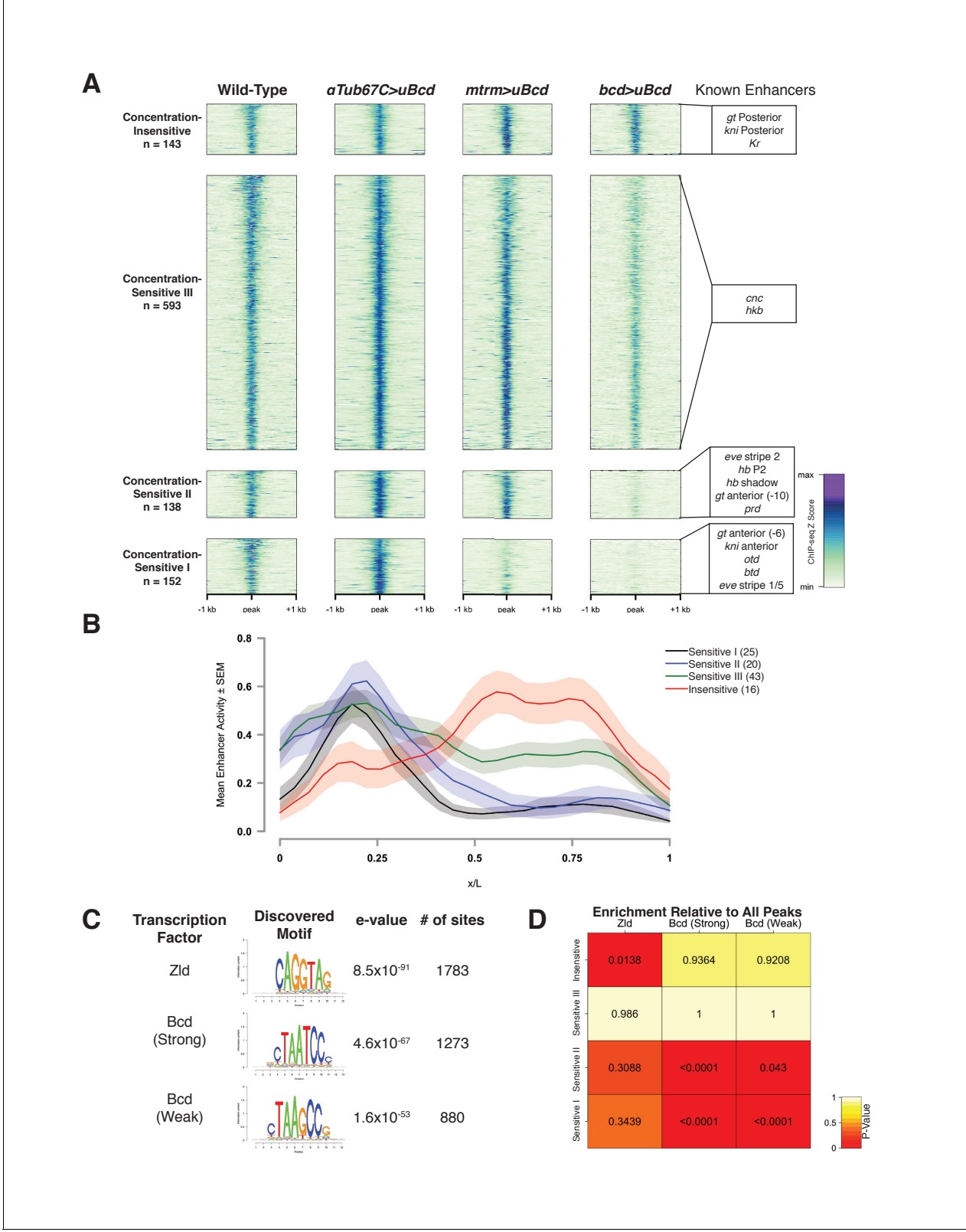

**Figure 2.** Bcd-bound regions are classified into groups of increasing sensitivity to Bcd concentration. (**A**) ChIP-seq data in Bcd-bound peaks. Data is displayed as a heatmap of z-score normalized ChIP-seq reads, in a 2 kilobase region centered around each peak. Peaks in each class are arranged in order of decreasing z-scores in wild-type embryos. One peak (peak 549, see **Supplementary file 1**) was not classified, as it showed increasing binding at decreasing Bcd concentrations. Previously characterized enhancers overlapping with each class are indicated at right. Concentration-Insensitive: the
*Figure 2 continued on next page*

*Figure 2 continued*

posterior stripe enhancers for both *knirps* (*Pankratz et al., 1992*) and *giant* (*Schroeder et al., 2004*), and the *Krüppel* CD1 enhancer (*Hoch et al., 1991*). Concentration-Sensitive III: *cap'n'collar* (*Schroeder et al., 2004*), and *huckebein* (*Häder et al., 2000*) enhancers. Concentration-Sensitive II: the *hunchback* P2 proximal (*Struhl et al., 1989*) and shadow enhancers (*Perry et al., 2011*), the *even-skipped* stripe 2 enhancer (*Goto et al., 1989*), an early *paired* enhancer (*Ochoa-Espinosa et al., 2005*), and an anterior enhancer for *giant* (*Schroeder et al., 2004*). Concentration-Sensitive I: *buttonhead* (*Wimmer et al., 1995*), *orthodenticle* (*Gao and Finkelstein, 1998*), and anterior enhancers for both *knirps* and *giant* (*Schroeder et al., 2004*). (B) Mean expression patterns of Vienna Tile-GAL4 enhancer reporters overlapping with Bcd peaks in each sensitivity class. Peaks and Vienna Tiles with more than one overlap, as well as 11 Vienna Tiles that drove expression at a level too low to quantify, were excluded from the plot. (C) Top DNA motifs discovered by RSAT peak-motifs. The e-value for is a p-value computed from a binomial distribution for a given motif in the dataset, corrected for multiple testing. See *Figure 2—figure supplement 1* for de novo motif discovery in each sensitivity class. (D) Heatmap displaying the enrichment of a given motif in each sensitivity class, relative to the peak list as a whole. P-values were generated from permutation tests (n = 10,000 tests).

DOI: https://doi.org/10.7554/eLife.28275.005

The following figure supplements are available for figure 2:

**Figure supplement 1.** Enrichment for binding and motifs of transcription factors in Bcd sensitivity classes.

DOI: https://doi.org/10.7554/eLife.28275.006

**Figure supplement 2.** In vitro binding affinity of target enhancers for Bcd protein is insufficient to explain in vivo binding behavior.

DOI: https://doi.org/10.7554/eLife.28275.007

Bcd targets, we queried the Fly Enhancer resource generated from the Vienna Tile GAL4 reporter library (*Kvon et al., 2014*). The Fly Enhancer collection is a library of candidate enhancer DNA fragments (Tiles) driving expression of GAL4 that covers 13.5% of the non-coding genome. Each Tile's expression pattern has been measured and scored by developmental stage. A total of 293 Tiles overlap with at least one peak in our data set. Of these, 163 drive gene expression in stage 4–6 (which includes NC14), and these active Tiles overlap with a total of 151 (14.7%) of the Bcd-peaks (*Table 2*). The remaining overlapping Tiles either are active later in development (75), or are not functional (55). The fraction of the 293 Tiles that overlap with Bcd peaks and are active during early development (163, or 55.6%) is significantly higher than the proportion of Tiles active in early embryos for the collection as a whole (666 of the total 7793 or 8.5%). This suggests that the Bcd binding peaks are significantly enriched for enhancer activity, and that a similar fraction of the remaining 793 Bcd peaks not present in the Fly Enhancer collection may correspond to active enhancers as well.

The proportion of peaks overlapping with active enhancers in the Fly Enhancer library varies by sensitivity class. While the Concentration-Sensitive III peaks are the largest class, they have the lowest representation in the Fly Enhancer collection, with 18% of peaks overlapping with a Vienna Tile

**Table 2.** Number of overlapping Bcd ChIP peaks and Vienna Tile-GAL4 enhancer reporters.
Note that some Bcd peaks overlap with more than one Vienna Tile, and vice versa. The reporters expressed at stage 4–6 that overlapped with more than one Bcd peak were excluded from the plot in *Figure 2B*.

**Overlaps with Vienna Tile-GAL4 reporters**

| | | Total overlaps | | |
| --- | --- | --- | --- | --- |
| | Total Vienna tiles | Vienna tiles | Bcd peaks | Single overlaps |
| Active (all stages) | 3604 | 238 | 193 | 127 |
| Active (stage 4–6) | 666 | 163 | 151 | 115 |
| Patterned (stage 4–6) | 627* | 159 | 147 | 112 |
| Not Active | 4189 | 55 | 41 | 28 |
| Total | 7793 | 293 | 234 | 155 |

*The patterned expression from the 666 Vienna Tiles that drive expression at stage 4–6 was determined by subtracting the number of tiles scored as 'ubiquitous' (39) from the total number of tiles.

DOI: https://doi.org/10.7554/eLife.28275.008

enhancer reporter. Of these overlapping peaks, 52% overlap with Tiles active at stage 4–6, a smaller fraction than observed for the other peak classes which range from 73% to 77% (see *Table 3*). We also observe differences in the proximity of peaks in each class to transcription start sites (TSS), with the Concentration Sensitive III and Concentration-Insensitive classes being nearly twice as likely to lie within 500 bp of a TSS (see *Table 4*). Although highly accessible regions such as promoters are common artifacts in ChIP-seq data (*Teytelman et al., 2013*), we made substantial efforts to remove these artifacts (see Materials and methods). It is possible that the large Concentration-Sensitive III class contains peaks where Bcd binds opportunistically and its binding does not necessarily result in active gene expression, as has been observed in ChIP-seq datasets (*Biggin, 2011*). However, the large overlap between active enhancers in the Fly Enhancer database and our peaks list suggests that every class contains functional enhancers beyond those that have been previously examined.

The Bcd sensitivity classes are predictive of the expression domains of associated enhancer fragments. Enhancers overlapping with both the Concentration-Sensitive I and II classes drive expression in anterior regions of the embryo, with the Concentration-Sensitive III and Concentration-Insensitive classes driving broad and posterior expression, respectively (*Figure 2B*). This indicates that our classifications of the Bcd-bound peaks reflect unique groups of Bcd targets with differing abilities to bind Bcd protein and consequently activate gene expression in different positions along the AP axis.

## Sequence composition of ChIP sensitivity classes does not account for in vivo sensitivity to bcd concentration

We next wanted to determine whether the Bcd-bound regions in each sensitivity class differ at the level of DNA sequence. In vitro, Bcd binds with high affinity to the consensus 5'-TCTAATCCC-3', and that variations on this consensus sequence constitute weak binding sites (*Burz et al., 1998*; *Driever and Nüsslein-Volhard, 1989*; *Driever et al., 1989*). If the affinity of a given enhancer for Bcd were encoded primarily at the level of its DNA sequence, we would expect to see a higher representation of strong Bcd binding sites in the less sensitive classes, and weaker sites in the more sensitive classes. To test this, we performed de novo motif discovery using the RSAT peak-motifs algorithm (*Thomas-Chollier et al., 2012*; *2008*). We identified the top motifs in the entire Bcd ChIP peak list, ranked by their e-value, and found that the top three most highly ranked motifs were the consensus binding site for the proposed pioneer factor Zelda (Zld) (*ten Bosch et al., 2006*; *De Renzis et al., 2007*; *Harrison et al., 2011*; *Nien et al., 2011*), and a strong (TAATCC) and weak (TAAGCC) Bcd binding site (*Figure 2C* and *Figure 2—figure supplement 1*). We next calculated the frequency with which these motifs appear in each peak, and tested for enrichment between sensitivity classes by permutation test (*Figure 2D*). We found that despite their failure to bind Bcd at low concentrations, the Concentration-Sensitive I and II classes are enriched for both the strong and weak Bcd sites relative to the peak set as a whole. The Concentration-Sensitive III class did not contain an enrichment of any site over the total peak set. The Concentration-Insensitive class, however, showed a higher prevalence of the Zld binding site relative to the total peak set than any other class.

*Ochoa-Espinosa et al. (2005)* found little correlation between either the number or strength of Bcd binding sites in an enhancer and the position of gene expression driven by that enhancer. Given our result that the Concentration-Sensitive I and II classes drive expression primarily in the anterior

**Table 3.** Number of Bcd ChIP-seq peaks in each class overlapping with Vienna Tile-GAL4 reporters. Queried Bcd peaks are peaks that are present in the Fly Enhancer database, whose expression can be assessed.

**Overlaps of peak classes with active enhancers and transcription start sites**

| Bcd peak class | Overlaps with Vienna tiles | | Overlaps with Vienna tiles active at stg. 4–6 | | | Total bcd peaks |
| | N | % of Total Bcd Peaks | N | % of Queried Bcd Peaks | % of Total Bcd Peaks | |
| --- | --- | --- | --- | --- | --- | --- |
| Concentration-Sensitive I | 52 | 34.2% | 38 | 73.1% | 25.0% | 152 |
| Concentration-Sensitive II | 38 | 27.5% | 29 | 73.7% | 21.0% | 138 |
| Concentration-Sensitive III | 109 | 18.4% | 57 | 52.3% | 9.6% | 593 |
| Concentration-Insensitive | 34 | 23.8% | 26 | 76.5% | 18.2% | 143 |

DOI: https://doi.org/10.7554/eLife.28275.009

**Table 4.** Number of Bcd ChIP-seq peaks in each class within 500 bp Transcription Start Sites.

| Bcd peak class | Overlaps with TSS | | Total bcd peaks |
|---|---|---|---|
| | N | % | |
| Concentration-Sensitive I | 29 | 19.1% | 152 |
| Concentration-Sensitive II | 35 | 25.4% | 138 |
| Concentration-Sensitive III | 291 | 49.1% | 593 |
| Concentration-Insensitive | 74 | 51.7% | 143 |

DOI: https://doi.org/10.7554/eLife.28275.010

of the embryo (*Figure 2B*), the higher density of Bcd binding sites in these peaks indicates that there is, in fact, a correlation between the binding site number and position of gene expression. This difference likely reflects the larger sample size used in our study, as well as our method for classifying Bcd bound peaks. However, in contrast to a binding site affinity model for Bcd function, Bcd targets that behave as concentration-sensitive and -insensitive in vivo are not distinguished by their representation of strong versus weak Bcd binding sites, partially confirming the previous study (*Ochoa-Espinosa et al., 2005*).

In further support of this concept, we found little correlation between in vitro binding affinity by electrophoretic mobility shift assay and the in vivo binding properties we observe by ChIP for a selected subset of peaks (*Figure 2—figure supplement 2*). At the level of sequence composition, they instead appear to differ in their balance of Bcd and Zld binding sites. Although both strong and weak Bcd sites and Zld sites are enriched in the Bcd ChIP peaks as a whole, there is a bias toward both Bcd sites in peaks that show concentration-sensitive binding properties by ChIP-seq and a bias toward Zld sites in the concentration-insensitive peaks. Zld, a ubiquitously expressed early embryonic transcription factor, has been implicated in chromatin remodeling prior to zygotic genome activation (*Harrison et al., 2011*; *Nien et al., 2011*; *Sun et al., 2015*). The predominance of Zld motifs over Bcd motifs in the Concentration-Insensitive class suggests that in vivo chromatin structure also plays a role in the sensitivity of a given target to transcription factor concentration in the context of the developing embryo. Taken together, these findings suggest that the chromatin context of an enhancer may play a greater role in its overall affinity for a transcription factor in vivo than the sequences of the binding sites that it contains. We therefore set out to test the hypothesis that sensitivity classes are distinguished at the level of chromatin structure.

## Bcd is required for chromatin accessibility at a subset of concentration-sensitive target sites

To measure genome-wide patterns of chromatin accessibility and nucleosome positioning, we performed ATAC-seq (*Buenrostro et al., 2015*) on single wild-type embryos precisely staged at 12 min after the onset of NC14, and identified 13,266 peaks of chromatin accessibility (see Materials and methods). Of the 1,027 Bcd-bound regions identified by ChIP-seq, 855 (83.3%) of them overlap with ATAC-seq peaks.

Given Zelda's role in influencing chromatin accessibility and the presence of its binding sites at Bcd-bound regions of genome, we measured the effect of Zld on accessibility at Bcd sites by ATAC seq (*Figure 3A*). Of the total 13,226 accessible regions at NC14, 2675 (20.2%) show a significant reduction in accessibility in *zld* mutant embryos. This fraction is higher in Bcd-bound peaks; 402 (39.1%; or 379 [44.3%] of the 855 Bcd peaks that overlap with ATAC open regions, see *Table 5*) show reduced accessibility in *zld* mutants, indicating that Bcd bound regions are more likely to be dependent on Zld for their accessibility than the genome as a whole. However, the Zld-dependent peaks are distributed across each sensitivity class determined by ChIP, with no particular class being significantly more Zld-dependent (*Figure 3B*). This contrasts with the distribution of binding sites in the peak classes, which revealed that the Concentration-Insensitive peaks were more likely to contain Zld binding sites. However, if the Bcd peaks are separated into those bound and not bound by Zld, we do observe an enrichment of the Concentration-Insensitive peaks in Zld-dependent, Zld-bound peaks (*Figure 3—figure supplement 1*). These results suggest that while Zld contributes to the

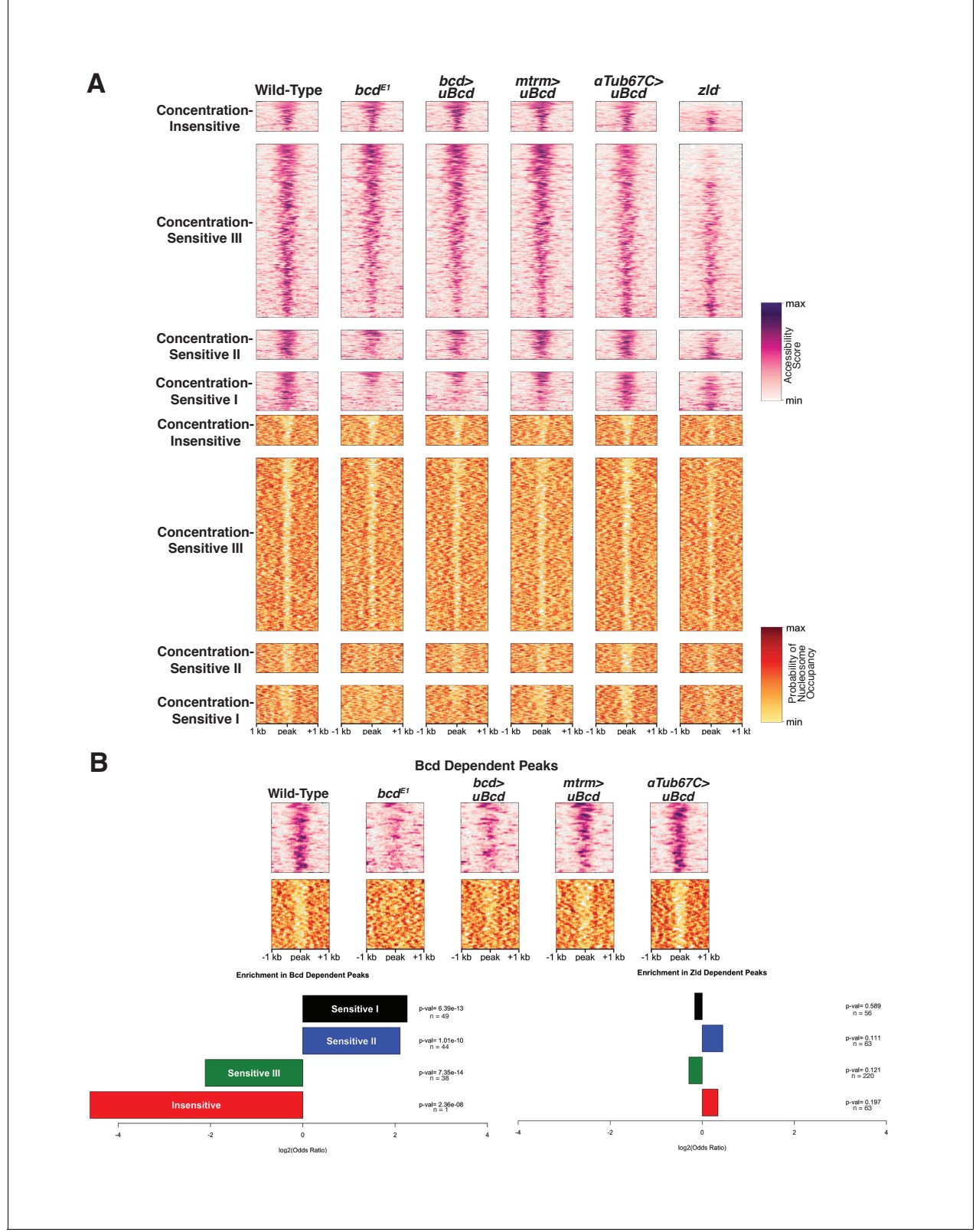

**Figure 3.** Bcd drives chromatin accessibility primarily at concentration-sensitive targets. (**A**) Heatmaps showing chromatin accessibility (top) and probability of nucleosome occupancy (bottom) around Bcd-bound peaks from ATAC-seq experiments. Peak regions are arranged by decreasing accessibility in wild-type embryos. *bcd^E1* mutant embryos show a loss of accessibility and increased nucleosome occupancy most strongly at the Concentration-Sensitive I and II peaks. *zld^-* embryos show reduced accessibility across all sensitivity classes. (**B**) Subset of 132 Bcd-bound peaks selected

*Figure 3 continued on next page*

*Figure 3 continued*

from (**A**) that become inaccessible in the absence of Bcd. Accessibility at these peaks increases with increasing concentrations of uniform Bcd. Odds ratios and p-values calculated from Fisher's exact test show significant overrepresentation of the Concentration-Sensitive I and II classes in the Bcd- and Zld-dependent peaks. n values indicate the number of peaks in each sensitivity class that are Bcd or Zld dependent.

DOI: https://doi.org/10.7554/eLife.28275.012

The following figure supplement is available for figure 3:

**Figure supplement 1.** Enrichment of Bcd sensitivity classes in Zld bound peaks.

DOI: https://doi.org/10.7554/eLife.28275.013

accessibility of a subset of Bcd targets, it is unlikely to be the sole determinant of the differential concentration sensitivity of Bcd peaks as a whole.

Given the enrichment for both strong and weak Bcd binding sites in the Concentration-Sensitive I and II classes, we next examined the impact of Bcd protein itself on chromatin accessibility by ATAC-seq (*Figure 3A*). In *bcd* mutants, 326 (2.4%) of the 13,266 open regions in wild-type embryos show significantly reduced accessibility accompanied by increased nucleosome occupancy in those same regions (*Figure 3A and B*). These regions are therefore either directly or indirectly dependent on Bcd for their accessibility. More strikingly, 132 (12.9%) of the 1,027 Bcd ChIP-seq peaks show reduced accessibility in the absence of Bcd and likely represent regions where Bcd's impact is direct. These regions dependent on Bcd for accessibility are significantly enriched for peaks in the Concentration-Sensitive I and II classes (32.9% and 31.9% of each class, with Fisher's exact test P-values of $4.29 \times 10^{-12}$ and $1.37 \times 10^{-10}$, respectively). In contrast, the Concentration-Sensitive III and Concentration-Insensitive classes are both significantly underrepresented (6.07% and 0.7% and p-values=$7.88 \times 10^{-13}$ and $2.29 \times 10^{-8}$) (*Figure 3B*). This suggests that Bcd binding influences chromatin accessibility preferentially at a subset of highly concentration-sensitive targets.

Because the Concentration-Sensitive I and II classes are bound primarily at high Bcd concentrations, Bcd's effects on chromatin accessibility at these targets likely occurs only in anterior regions of the embryo. In support of this, we find that chromatin accessibility at Bcd-dependent, concentration-sensitive targets is responsive to Bcd concentration. Expressing uniform Bcd confers accessibility to peaks that are not accessible in *bcd* mutant embryos (*Figure 3B*). The degree of chromatin accessibility conferred by Bcd correlates positively with the concentration of uniform Bcd expressed (*Figure 3B*). This observation, along with the overrepresentation of the Concentration-Sensitive I and II classes in the Bcd-dependent peaks, suggests that Bcd influences the chromatin state of these targets primarily at the high concentrations found in the anterior of the embryo.

A second feature that distinguishes the chromatin structure of Bcd binding sites is the presence of DNA sequences favorable for nucleosome occupancy (*Segal et al., 2006*). Bcd bound regions in wild-type embryos are generally depleted of nucleosomes (*Figure 4A*). However, predicting nucleosome positioning sequences using the NuPoP algorithm (*Xi et al., 2010*) suggests that the Concentration-Sensitive I and II Bcd peak classes are more likely to bind nucleosomes than the Concentration-Sensitive III and Concentration-Insensitive classes. (*Figure 4B*). The contrast between predicted occupancy and observed depletion suggests that these regions are actively restructured for Bcd and other transcription factors to bind. The increased nucleosome preference of the more Concentration-Sensitive peaks, combined with the observation that these sites become occupied by nucleosomes in *bcd* mutants suggests a model where Bcd, either directly or in combination with

**Table 5.** Number of peaks dependent on Bcd or Zld for chromatin accessibility.

ATAC +ChIP common peaks are peaks that overlap between the Bcd ChIP-seq peaks and the wild-type ATAC-seq open chromatin peaks.

| Peak list | Bcd Dependent | Zld Dependent | Bcd + Zld dependent | Total |
|---|---|---|---|---|
| ATAC Open Peaks | 326 | 2675 | 206 | 13,226 |
| Bcd ChIP Peaks | 132 | 402 | 61 | 1027 |
| ATAC + ChIP Common Peaks | 121 | 379 | 58 | 855 |

DOI: https://doi.org/10.7554/eLife.28275.011

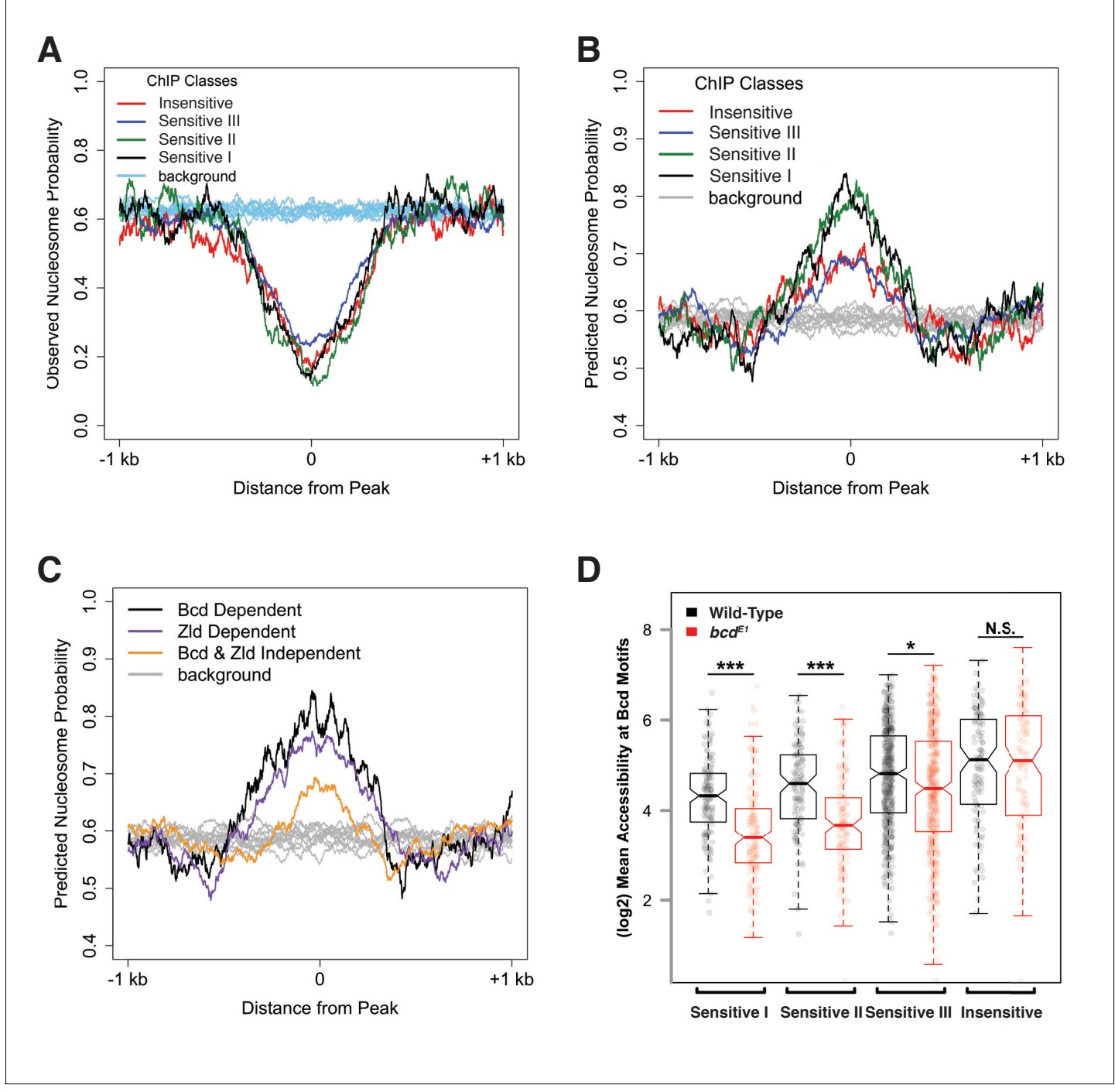

**Figure 4.** Bcd sensitivity classes differ in both predicted and observed nucleosome occupancy. (A) Metaprofiles of nucleosome occupancy in each sensitivity class in wild-type embryos. Background represents random selection of regions outside of Bcd peaks shows a genome-wide average nucleosome probability of ~0.6. Bcd-bound peak regions show reduced nucleosome occupancy compared to unbound regions. (B) Predicted nucleosome occupancy using NuPoP show higher modeled probability of nucleosome occupancy in Bcd-bound peaks relative to background regions, with higher probability of occupancy at the Concentration-Sensitive I and II classes. (C) Predicted nucleosome occupancy in peaks dependent on Bcd vs. Zld ($n_{Bcd}$ = 132 peaks, $n_{Zld}$ = 402 peaks, with n = 61 peaks dependent on both Bcd and Zld) for accessibility show higher predicted occupancy than peaks independent of both Bcd and Zld (n = 554). (D) Mean wild-type (black) or $bcd^-$ (red) ATAC accessibility scores for Bcd motifs were calculated for each peak and plotted by sensitivity group. Boxplots depict the distribution of accessibility scores for each group in each genotype, and individual data points are shown as points. P-values were calculated by one-sided permutation test and indicate the likelihood in a randomly selected population of observing a difference between means greater than the observed values (p<1e-6 for Concentration-Sensitive I and II groups, p=0.001207 for Concentration-Sensitive III, and p=0.988167 for Concentration-Insensitive).

*Figure 4 continued on next page*

*Figure 4 continued*

DOI: https://doi.org/10.7554/eLife.28275.014

cofactors is able to direct chromatin remodeling events, which may play a significant role in distinguishing concentration-sensitive and -insensitive targets. Additionally, we find that Bcd-bound regions that are dependent on either Zld or Bcd for their accessibility are more likely to have a higher nucleosome preference than regions that are independent of both factors (*Figure 4C*). This further suggests that Bcd is able to overcome a high nucleosome barrier in a manner similar to Zld (*Sun et al., 2015*) at a subset of its target enhancers.

These effects of chromatin accessibility impact the availability of sequence motifs for binding Bcd. In wild-type embryos, there is a gradual increase in average motif accessibility from high to low sensitivity, and this difference becomes more pronounced in *bcd* mutant embryos (*Figure 4D*), consistent with a role for Bcd in driving changes in accessibility at more sensitive sites in a concentration dependent manner. This is also evident at the level of nucleosome organization. Calculating the fraction of motifs that overlap nucleosomes in either wild-type or *bcd* mutant chromatin conformations, we find that whereas on average across all sensitivity classes $55 \pm 2\%$ of Bcd motifs are in nucleosome-free tracts in wild-type embryos, in *bcd* mutant embryos motifs have lower overall accessibility and a graded association with nucleosomes that correlates with the sensitivity classes (41%, 46%, 50%, and 53% of motifs are accessible from high to low sensitivity). These results indicate that the mechanistic determinants of concentration-dependent Bcd action likely involve a complex interaction between Bcd, DNA, and chromatin structure.

## A truncated bcd protein shows reduced binding specifically at concentration-sensitive targets

A chromatin remodeling activity associated with Bcd has not been previously described. We hypothesize that Bcd renders its target sites accessible either by competing with nucleosomes to access its binding sites and bind to DNA at high concentrations or by recruiting chromatin-remodeling enzymes to accessible motifs and subsequently driving local nucleosome remodeling to render more sites accessible. We reasoned that if Bcd can displace nucleosomes simply by competing with them for access to its binding sites, it should be possible for the Bcd DNA-binding homeodomain to compete. However, if Bcd instead drives remodeling via recruitment of cofactors, it is likely that these interactions or activities are carried out through regions of the protein outside of the DNA binding domain. To distinguish between these two possibilities, we designed a transgenic GFP-Bcd construct that is truncated downstream of the homeodomain. We modeled the truncated Bcd protein after the *bcd*[085] allele, which was originally classified as an 'intermediate allele' of *bcd* (*Frohnhöfer and Nüsslein-Volhard, 1986*) and reported to have weak transcriptional activating activity (*Struhl et al., 1989*). The truncation occurs 28 amino acids downstream of the homeodomain (*Figure 5A*), and the GFP-tagged protein was therefore expected to bind DNA but lack functions requiring its C-terminus. The truncated protein (known as GFP-Bcd[085]) forms a gradient from the anterior of the embryo, and is expressed at a similar level as a full-length GFP-Bcd (*Figure 5B*).

While it is difficult to determine whether the homedomain structure is completely intact in the GFP-Bcd[085] construct, we tested the ability to the truncated protein to bind to Bcd target sequences compared to a wild-type GFP-Bcd protein. By ChIP-seq, we found that GFP-Bcd[085] binding in the Concentration-Sensitive I and II peak classes was significantly reduced compared to wild-type (p-values<0.0001 in permutation test with n = 10,000 trials), while binding to the Concentration-Sensitive III and Concentration-Insensitive classes was not (*Figure 5C*). Our ATAC-seq experiments revealed that these classes have reduced chromatin accessibility in *bcd* mutants (*Figures 3B* and *4D*). Taken together, these results suggest that Bcd's ability to access its concentration-sensitive targets is dependent upon activities carried out by domains in the C-terminus of the protein, likely via recruitment of a cofactor, and that its DNA binding activity alone is insufficient to drive chromatin accessibility.

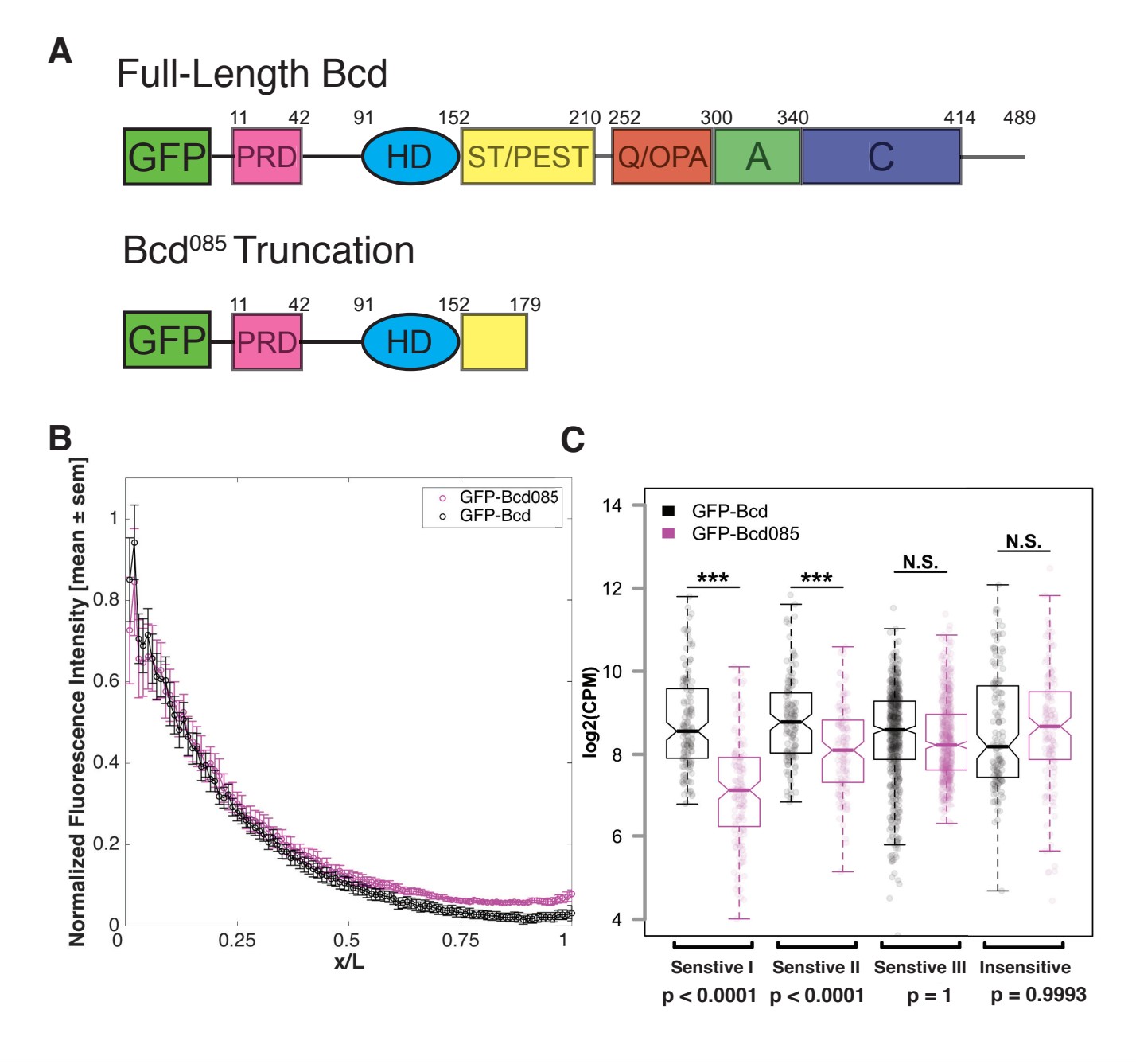

**Figure 5.** Bcd requires C-terminal protein domains to bind to concentration-sensitive targets. (**A**) GFP-Bcd[085] construct is truncated within the S/T domain downstream of the homedomain. Wild-type protein domains modified from (*Janody et al., 2001*) and (*Crauk and Dostatni, 2005*). The N-terminus of the protein includes a PRD repeat, followed by the DNA-binding homeodomain (HD) (*Berleth et al., 1988*). The serine/threonine-rich (S/T) domain is the target of MAPK phosphorylation by the terminal patterning Torso pathway (*Janody et al., 2000*) and contains a PEST sequence implicated in targeting the protein for degradation (Rechsteiner and Rogers, 1996). The C-terminus contains three domains implicated in transcriptional activation. The glutamine-rich (**Q**)/OPA and alanine-rich (**A**) domains are required for interactions with TAFII110 and TAFII60, respectively (*Sauer et al., 1995*). The acidic (**C**) domain has been demonstrated to play a role in transcriptional activation in yeast, but is not required for Bcd activity in the embryo (*Driever et al., 1989*). (**B**) GFP-Bcd[085] forms a protein gradient comparable to wild-type GFP-Bcd. GFP fluorescence intensity was extracted from dorsal profiles of live embryos. Error bars are standard error of the mean: GFP-Bcd embryos, n = 8; and GFP-Bcd[085] embryos, n = 8. (**C**) Boxplots displaying log transformed CPM normalized ChIP-seq data from *GFP-Bcd;;bcd[E1]* (wild-type) and *GFP-Bcd[085];bcd[E1]* (Bcd[085]) embryos show significant reduction binding of Bcd[085] in Concentration-Sensitive I and II peaks. P-values were calculated from permutation tests (n = 10,000).

DOI: https://doi.org/10.7554/eLife.28275.015

## Bicoid binding sites confer anterior expression to a posterior target

Overall, highly concentration-dependent targets are expressed in the anterior and are dependent on Bcd for accessibility, while less sensitive targets show more posterior expression patterns and a greater enrichment for Zld binding sites. An enhancer for *caudal* is a Concentration-Insensitive Bcd target and drives expression in the posterior of the embryo (*Figure 6*). This enhancer depends on Zld for chromatin accessibility, and consequently is not functional in *zld* mutants (*Supplementary file 1* and *Figure 6B*). This supports previous findings that Zld binding contributes to allowing Bcd activation at low concentrations in posterior nuclei (*Xu et al., 2014*). Like the *kni* posterior enhancer (*Figure 1A*), the *caudal* enhancer is Bcd independent for chromatin accessibility and its expression boundary shifts anteriorly in *bcd* mutant embryos. We tested whether we could convert the properties of the *caudal* enhancer from low to high sensitivity by manipulating DNA motifs. We identified the Zld binding sites in the *caudal* enhancer sequence and mutated them to Bcd binding sites (*Figure 6A*). These mutations result in a shift of *caudal* reporter expression to the anterior of the embryo. Anterior expression of the mutated reporter is Bcd dependent, as it is lost in *bcd* mutant embryos. Importantly, the mutant enhancer is functional in *zld* mutant embryos, retaining a distinct anterior expression domain. In the absence of Zld binding, the wild-type enhancer does not drive expression (*Figure 6B* and *Figure 6—figure supplement 1A*). By replacing Zld motifs with Bcd motifs, the enhancer retains functionality, but the spatial domains of expression are now restricted to regions of high Bcd concentration. While technical limitations prevented us from measuring chromatin accessibility of the reporter constructs directly, these results are consistent with a model where Bcd operates at high concentrations to confer chromatin accessibility at target sites. Previous work has demonstrated that addition of Zld sites to an inactive DNA fragment (HC_45) can transform it into an enhancer that drives gene expression early in development (*Xu et al., 2014*). We found that adding Bcd sites instead of Zld sites to this same reporter also activates expression from the enhancer in an anterior stripe (*Figure 6—figure supplement 1B*). These data support a model where Bcd delineates distinct gene expression and chromatin states at specific positions along its concentration gradient.

## Discussion

### A model for chromatin accessibility thresholds at bcd target genes

The results presented here demonstrate that the positional information in the Bcd gradient is read out as differential binding between Bcd and the *cis*-regulatory regions of its target genes. The over-representation of enhancers for anteriorly expressed target genes in the more sensitive classes provides support for the classic French flag model, as their enhancers are only capable of binding Bcd at high levels. However, motif analysis and in vitro EMSA experiments reveal that the differences in binding affinities that we observe in vivo cannot be explained entirely by the sequence of Bcd binding sites in the target enhancers. Instead we find that a subset of the enhancers in the concentration-sensitive classes require Bcd for chromatin accessibility. Taken together, this leads us to model in which the Bcd morphogen establishes concentration thresholds along the AP axis of the developing embryo by driving opening chromatin states at high concentrations, thereby gaining access to its most sensitive target enhancers. At lower concentrations in more posterior nuclei, Bcd is unable to access these enhancers, and therefore does not bind and activate their transcription (*Figure 7*). In this way, expression of these concentration-sensitive target genes is restricted to anterior regions of the embryo. The higher density of Bcd binding sites in highly concentration-sensitive target enhancers (shown in *Figure 2D*) suggests that these targets may require a larger number of Bcd molecules to be bound at a given time to keep them free of nucleosomes and accessible to the additional regulatory factors. We therefore provide a model for morphogen function in which the concentration thresholds in the gradient are read out molecularly at the level of chromatin accessibility, rather than through the strength of binding sites in the target sequences.

It is important to note that the discrete sensitivity classes described here were generated by Bcd binding data, and this binding is expected to occur prior to the activation of target genes and refinement of their expression domains. In our model Bcd establishes these initial patterns not by competing with its own target genes, but with default nucleosome positions in the early embryo. We predict that this initial interaction with chromatin is an essential event for establishing distinct,

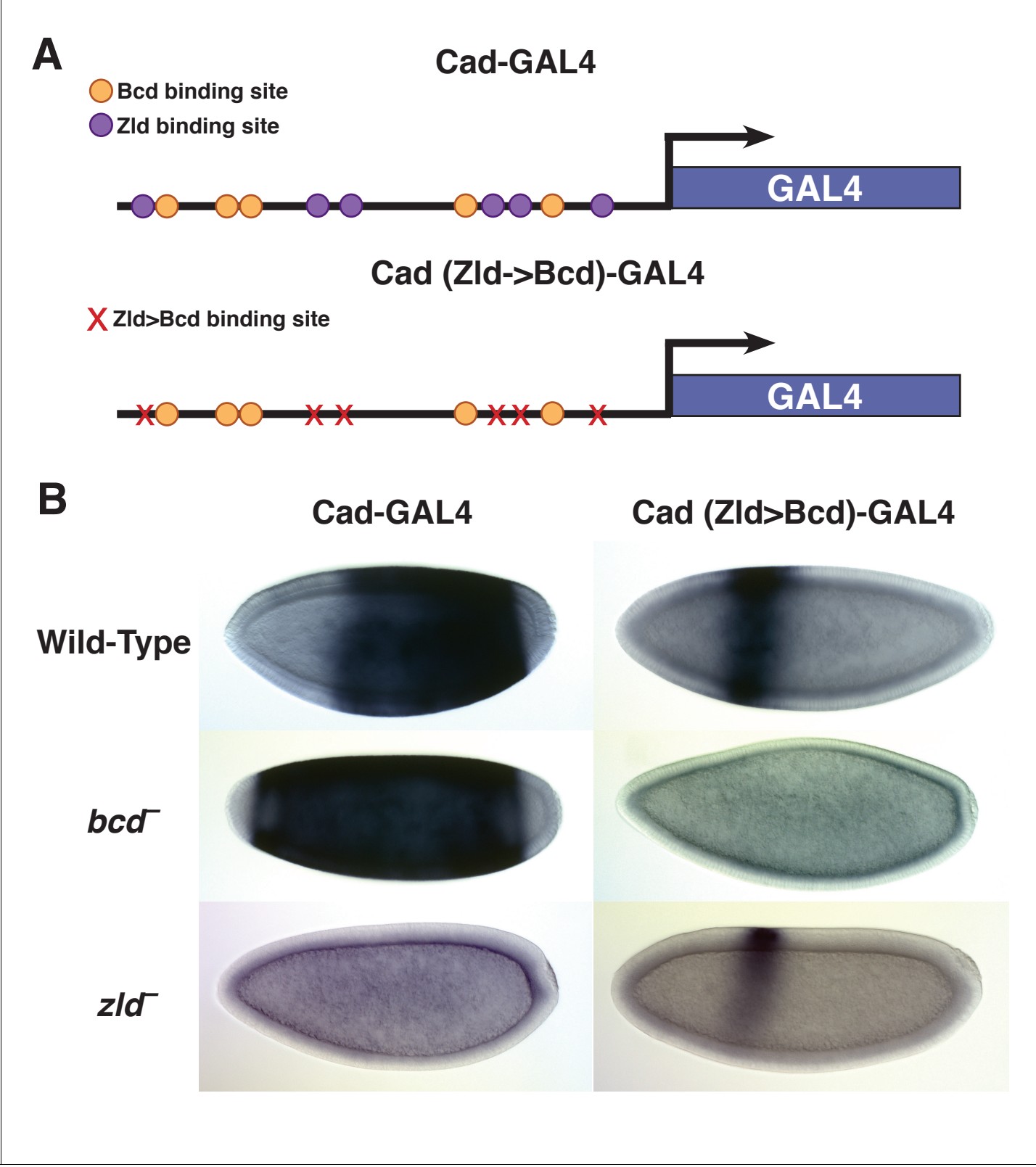

**Figure 6.** Replacing Zld sites with Bcd sites shifts gene expression to the anterior. (A) Schematic of the Vienna Tile enhancer reporter for *caudal*, containing 5 Zld and 6 Bcd binding sites. The mutated reporter contains 11 Bcd binding sites and no Zld sites. (B) Expression of the wild-type and mutated reporter in wild-type, *bcd⁻* or *zld⁻* embryos.

DOI: https://doi.org/10.7554/eLife.28275.016

*Figure 6 continued on next page*

*Figure 6 continued*

The following figure supplement is available for figure 6:

**Figure supplement 1.** Enhancer reporter constructs with variable Bcd and Zld binding sites.

DOI: https://doi.org/10.7554/eLife.28275.017

positionally defined patterns of gene expression. The chromatin landscapes established early by Bcd are then elaborated upon by additional patterning factors, including Bcd target genes themselves, as well as the repressor gradients proposed by Chen, *et al.* (*Chen et al., 2012*) Thus, the pre-transcriptional information presented by Bcd in the form of differential binding states is refined at the level of gene expression domains both by Bcd and other transcription factors active in the early embryo.

## Relationship of bicoid and Zelda at Bicoid-bound enhancers

The prominence of the Zld binding motif in the Bcd-bound ChIP peaks and ATAC-seq in *zld* mutants reveals that Zld also contributes to the accessibility of Bcd targets in the genome, in part at those targets that are not dependent on Bcd for their accessibility (61/1,027 peaks are dependent on both Bcd and Zld for accessibility). Zld is therefore likely to be one component that influences the accessibility and therefore the apparent in vivo affinity of the peaks that are bound by Bcd but insensitive to its local concentration (*Figure 7*). Indeed, it has been demonstrated that Zld can extend the effective range of the dorsal-ventral morphogen Dorsal and aid in establishing concentration thresholds and interpreting the gradient at low concentrations, while the requirement for Zld is less significant at high concentrations (*Nien et al., 2011*). Previous work has also suggested that Zld contributes to activation of target genes at low concentrations of Bcd protein (*Xu et al., 2014*), and a recent study demonstrated optically that in the absence of Zld, Bcd occupancy on chromatin in the posterior of the embryo is reduced (*Mir et al., 2017*).

These studies, in combination with the work presented here, allow us to predict that transforming a concentration-sensitive Bcd target enhancer into a Zld dependent enhancer would increase the accessibility and therefore the sensitivity of that region in vivo. Indeed, Xu, *et al.* have previously demonstrated that adding increasing numbers of Zld sites to an inactive Bcd-bound enhancer can drive increasingly posterior gene expression (*Xu et al., 2014*). The reporter construct used to demonstrate this effect (HC_45) is identified as a Concentration-Sensitive I target in our study. We posit that the increase in gene expression from this reporter observed in their work is the result of increasing the accessibility of the enhancer region. Interestingly, adding four additional Bcd sites to this reporter also activates anterior gene expression. This suggests that at high concentrations, Bcd's activity is similar to Zld.

Alternately, when we replace Zld sites with Bcd sites in an enhancer that drives posterior expression, the expression domain shifts to the anterior of the embryo. This demonstrates that without Zld to keep the enhancer open in posterior nuclei, activation of the reporter gene becomes entirely dependent on Bcd, effectively shifting this reporter from a concentration-insensitive to concentration-sensitive enhancer. This finding fits with both previously reported findings and the model proposed in our study. Namely, that Zld contributes to the accessibility of Bcd target genes throughout the embryo, while high levels of Bcd can drive accessibility independently and activate gene expression at a subset of targets in the anterior of the embryo.

The reduced binding by a truncated Bcd protein at the most concentration-sensitive targets indicates that Bcd does not displace nucleosomes by simply by competing for binding to genomic targets, but rather that the C-terminus of the Bcd protein is required for accessing its nucleosome-associated DNA targets. Previous work has shown that various domains of the Bcd protein are required for interactions with both co-activators and co-repressors. The N-terminus of Bcd is required for interactions with components of the Sin3A/HDAC repressor complex, and these interactions are proposed to play a role in reducing Bcd's transcriptional activation activity (*Zhao et al., 2003*; *Zhu et al., 2001*). Multiple Bcd domains, including the C-terminus, are required for interaction with CREB-binding protein (CBP), which has histone acetyltransferase activity (*Fu and Ma, 2005*; *Fu et al., 2004*). It is possible that in our truncated Bcd construct, this interaction with CBP is disrupted. As CBP is thought to play a role in increasing chromatin accessibility for transcription factors

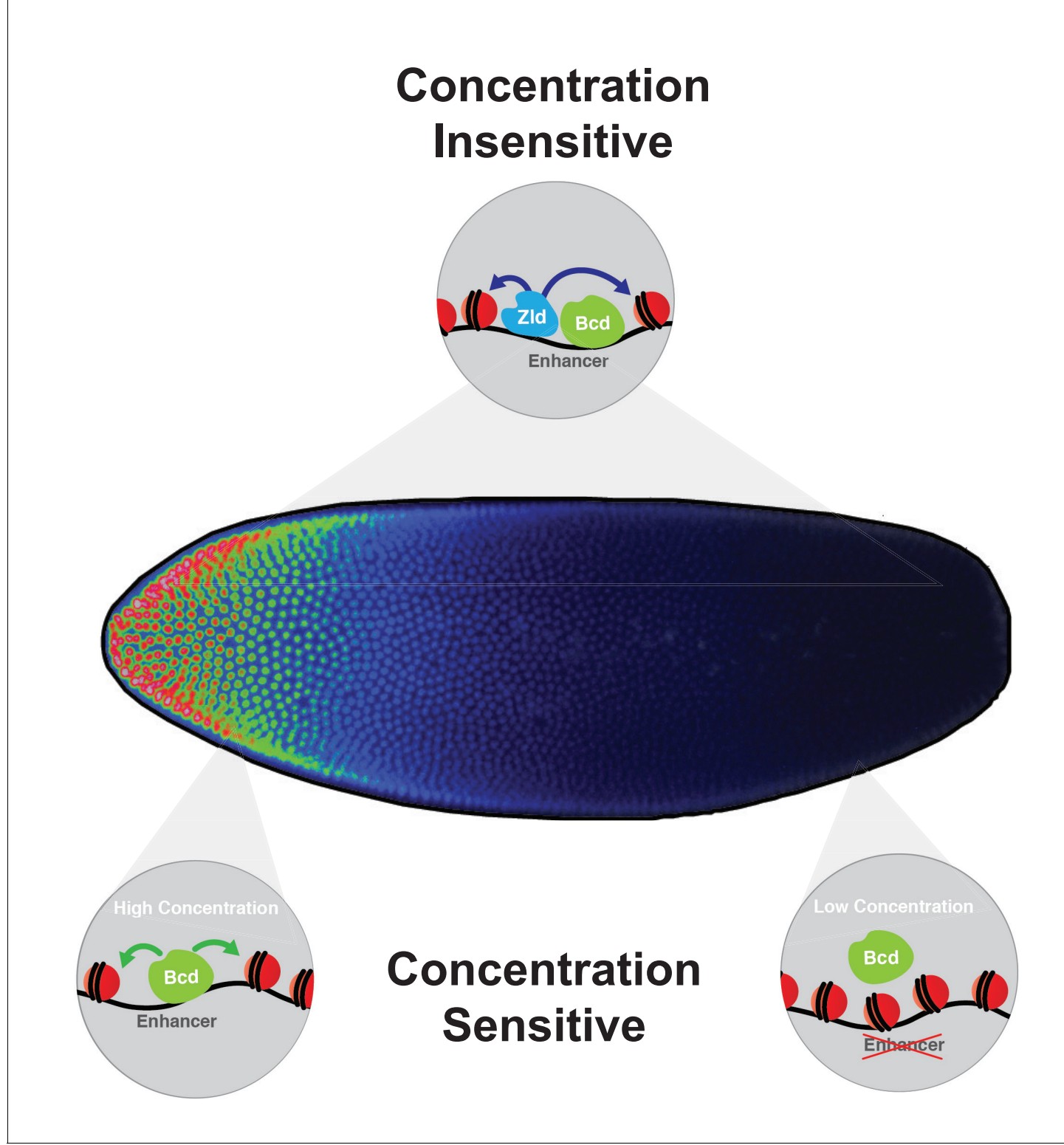

**Figure 7.** Model for Bcd function along the AP axis. Bcd drives accessibility of concentration-sensitive, Bcd-dependent enhancers at high concentrations in anterior nuclei, and these sites are closed in posterior nuclei. concentration-insensitive targets remain accessible in both anterior and posterior nuclei, likely through inputs from other factors such as Zld and more open local chromatin structure with a lower nucleosome preference.
DOI: https://doi.org/10.7554/eLife.28275.018

(*Chan and La Thangue, 2001*), the loss of this interaction could lead to the reduced binding to sensitive targets that we observe in embryos with truncated Bcd. The targets that we have classified as concentration-insensitive do not depend on Bcd to establish an open chromatin state. This suggests that these sites are opened by other chromatin remodeling factors, or are inherently more likely to be nucleosome-free based on their underlying sequence.

It has previously been suggested that transcription factors can compete with nucleosomes for access to their DNA binding sites (*Mirny, 2010*; *Wang et al., 2011*). This could occur through cooperative binding to nucleosome-associated enhancers: if one Bcd molecule could gain access to a binding site that was protected by a nucleosome, it could recruit additional Bcd protein molecules to bind to nearby sites and occlude nucleosome binding. This cooperativity would require a high concentration of Bcd protein, fitting with our observation that Bcd influences accessibility more strongly at high concentrations. However, our experiments with a truncated Bcd protein reveal that Bcd cannot bind to its most sensitive targets without its C-terminal domains. As many of the residues that have been implicated in cooperative binding reside in the Bcd homeodomain (*Burz and Hanes, 2001*), we would expect this truncated Bcd to bind cooperatively. This finding therefore supports a model in which Bcd is actively remodeling chromatin, either directly or more likely by interacting with chromatin remodeling factors through its C-terminus.

As a maternally supplied factor, Bcd provides one of the first cues to the break the symmetry of the embryonic body plan. Our results suggest that this symmetry breaking occurs first at the level of chromatin accessibility, as Bcd drives the opening of its most concentration-sensitive target enhancers in anterior nuclei. Another maternal factor, Zld, is proposed to act as a pioneer factor at early embryonic enhancers with a high intrinsic nucleosome barrier. By binding to these enhancers, Zld depletes them of nucleosomes and allows patterning transcription factors to bind and activate gene expression (*Sun et al., 2015*). We have demonstrated here that Bcd influences the accessibility primarily of its concentration-sensitive targets, which also exhibit a high predicted nucleosome barrier (*Figure 4B*). This raises the possibility that Bcd may be exhibiting pioneer-like activity at high concentrations, driving accessibility of these sites prior to transcriptional activation. It is unlikely that Bcd is unique in its ability to influence the local chromatin accessibility of its targets. Recent work in mouse embryos has shown that another homeodomain transcription factor, Cdx2, influences the chromatin accessibility of its targets during posterior axial elongation (*Amin et al., 2016*). It will therefore be of interest to assess whether other developmental transcription factors can drive accessibility of their genomic targets while the chromatin state of the genome is being remodeled during early development.

## Materials and methods

### Model organism

All reported experiments were performed on embryos from transgenic and mutant variants of the Fruit Fly *Drosophila melanogaster* (NCBI Taxon 7227).

### Fly stocks and Genetics

*bcd* mutants refers to embryos derived from *bcd*[E1] homozygous mothers. The *bcd*[E1] and *bcd*[E1] *nos*[L7] *tsl*[4] stocks were from the Wieschaus/Schüpbach stock collection maintained at Princeton University. *zld* mutants are embryos derived from *zelda*[294] germline clones. Zelda mutant embryos were generated from the *zld*[294] allele (kind gift of Christine Rushlow) as germline clones as described previously (*Blythe and Wieschaus, 2015*). Uniform Bcd and Bcd[085] transgenes were expressed in a *bcd*[E1] mutant background. Germline clones possessing only positional information from Bcd were generated by heat shocking *hsFLP; FRT82B hb*[FB] *nos*[BN] *tsl*[4]/*FRT82B tsl*[4] *Ovo*[D] larvae. Germline clones lacking Bcd positional information as well were generated by heat shocking *hsFLP; FRT82B bcd*[E1] *hb*[FB] *nos*[BN] *tsl*[4]/*FRT82B tsl*[4] *Ovo*[D] larvae. Embryos from homozygous *eGFP-Bcd; bcd*[E1] *nos*[L7] *tsl*[4] mothers were used in ChIP-seq experiments to determine the impact of removing other maternal factors on Bcd binding to its targets.

All ATAC-seq experiments were performed in a *His2Av-GFP* (Bloomington) background to facilitate scoring of nuclear density.

The uBcd transformants expressed eGFP-tagged Bcd in a graded distribution in the embryo and RFP in the eyes. Transgenic flies containing the uBcd constructs were crossed into a $bcd^{E1}$ background. To achieve uniform Bcd expression, the uBcd flies were crossed to a stock expressing a heat shock-inducible *flippase* in a $bcd^{E1}$ background and the resulting larvae were heat shocked at 37°C. Recombination of the FRT-flanked cassette containing the *bcd* 3'UTR and 3xP3-RFP was scored by a mosaic loss of RFP expression in the eyes. Mosaic flies were further outcrossed to $bcd^{E1}$ and progeny lacking the *bcd* 3'UTR were sorted by loss of RFP expression. The resulting flies produced embryos in which the *bcd* 3'UTR was replaced by the *sqh* 3'UTR causing a uniform distribution of Bcd along the AP axis. (**Figure 1—figure supplement 1C**)

## Transgenic constructs

The uniform Bcd constructs were generated using a pBABR plasmid containing an N-terminal GFP-tagged *bcd* cDNA in which the *bcd* 3'UTR was replaced by the *sqh* 3'UTR (pBABR GFP-Bcd3'sqh) (Oliver Grimm, unpublished). This results in a loss of mRNA localization at the anterior pole of the oocyte. A sequence containing the *bcd* 3'UTR and a 3xP3-RFP reporter flanked by FRT sites was synthesized by GenScript and cloned by Gibson Assembly into the pBABR GFP-Bcd3'sqh plasmid. The FRT-flanked cassette was inserted between the *bcd* coding sequence and the *sqh* 3'UTR. The *bcd* promoter was removed by digesting with AgeI and KpnI and replaced with either the *mtrm* or the *αTub67C* promoter to generate the *bcd-uBcd*, *mtrm-uBcd*, and *αTub67C-uBcd* constructs.

The GFP-Bcd[085] truncation was generated from eGFP-Bcd (**Gregor et al., 2007a**) in pBlueScript by amplifying with primers to create a stop codon after amino acid 179 as in the $bcd^{085}$ hypomorphic EMS allele (**Rivera-Pomar et al., 1996**). The primers inserted an AvrII restriction site 3' to the deletion site.

F Primer: 5'-TTGtag<u>CCTAGG</u>CCTGGATGAGAGGCGTGT-3'
R Primer: 5'-TCCAGG<u>CCTAGG</u>ctaCAAGCTGGGGGGATC-3'

The plasmid was amplified by PCR and the linear product was digested and ligated to create the Bcd[085] truncation. The GFP-Bcd[085] construct was digested from pBlueScript with BamHI and EcoRI and ligated into pBabr.

The uBcd and Bcd[085] constructs were injected for site directed transgenesis into embryos from a *y,w;attp40* stock by Genetic Services (*bcd-uBcd* and *αTub67C-uBcd*) or BestGene (*mtrm-uBcd* and Bcd[085]) and stable transformant lines were established. The mutant cad-GAL4 reporter was injected into a M{vas-int.Dm}ZH-2A, P{CaryP}attP2 stock by Rainbow Transgenic Flies, Inc.

The wild-type cad-GAL4 reporter (VT010589, coordinates chr2L: 20767347–20768825) was ordered from the Vienna Drosophila Resource Center (VDRC ID 205848/construct ID 210589). The mutated *cad* enhancer sequences were synthesized by GenScript, amplified using primers

F primer: 5'-<u>CACC</u>GGCACTCAGTAGAGCA-3'
R primer: 5'-GCGACCCAAAGACCAGAATA-3'

and inserted into the pBPGUw vector (**Pfeiffer et al., 2008**) by Gateway cloning using a pENTR/D-TOPO cloning kit (Thermo Fisher Cat # K240020).

The HC_45 and HC_45.4Z sequences were taken from **Xu et al. (2014)**. The HC45 +4 Bcd sequence was generated by replacing the extra Zld sites in HC45.4Z with Bcd binding sites. All three sequences were synthesized by GenScript, amplified using primers

F primer: 5'-<u>CACC</u>TCGATTTTCTGTTGCCATTTC-3'
R primer: 5'- CCTGATGGCTGTCCTCTGT-3'

and inserted into the pBPGUw vector (**Pfeiffer et al., 2008**) by Gateway cloning using a pENTR/D-TOPO cloning kit.

## Western blots

Live embryos were dechorionated in bleach, rinsed in salt solution (NaCl with TritonX-100), and embryos at NC14 were sorted under a light microscope and flash frozen on dry ice. Western blots were performed using a using a rabbit anti-GFP antibody (Millipore Cat # AB3080P) and mouse anti-tubulin antibody (Sigma Cat # T9026) as a loading control. For quantification, the GFP band intensities were normalized to α-tubulin band intensities in each lane. Two biological replicates of 50 embryos were homogenized in 50 µL buffer for each genotype, and 10 µL (=10 embryos) was loaded per lane.

Western blots were used to generate an estimate of Bcd concentration in each of the uniform Bcd lines. Drocco *et al.* used western blots to measure Bcd protein accumulation in the embryo during development, and calculated the total amount of Bcd in the embryo at NC14 to be $1.5 \pm 0.2 \times 10^8$ molecules (*Drocco et al., 2011*). Given that the volume of the nucleus is ~1/10 (or 1/1 + 9) the volume of the cytoplasm and Bcd partitions between the nucleus and the cytoplasm at a ratio of ~4:1 (*Gregor et al., 2007a*), we can generate a ratio of 4/4 + 9 or 0.31 for nuclear/cytoplasmic Bcd. Using this value, we can convert $1.5 \pm 0.2 \times 10^8$ molecules/embryo into $4.6 \times 10^7$ molecules/nucleus at NC14. At this stage, there are 6000 nuclei at the cortex of the embryo, which would be ~7,750 Bcd molecules/nucleus if the Bcd were distributed uniformly. Additionally, optical measurements estimate a nuclear concentration of Bcd as $8 \pm 1$ nM and 690 Bcd molecules at the *hunchback* expression boundary (~48% x/L) at NC14 (*Gregor et al., 2007b*). We used these values to generate a conversion factor of 0.011594203 nM/molecule and calculate the approximate nuclear concentrations given below (*Table 6*) for each uniform Bcd line. See also *Figure 1—figure supplement 1D*.

## Immunostaining and imaging

Embryos of indicated genotypes were collected from 0 to 4 hr laying cages, and fixed and stained essentially as described in (*Dubuis et al., 2013*), with rabbit anti-Bcd, guinea pig anti-Kni, and rat anti-Btd primary antibodies, followed by fluorophore-conjugated secondary antibodies Alexa-488 (guinea pig), Alexa-568 (rat), and Alexa-647 (rabbit) from Invitrogen. Stained embryos were imaged on a Leica SP5 laser-scanning confocal microscope.

## Live imaging and image analysis

Dechorionated embryos of the indicated genotypes were mounted on coverslips overlaid with halocarbon oil and imaged in the mid-sagittal plane on a Leica SP5 laser scanning confocal microscope. Image analysis was performed in MATLAB (http://www.mathworks.com). GFP intensity along the dorsal profile of each embryo was extracted for each frame of the live movies in nuclear cycle 14. The frame with the highest overall intensity in each movie was plotted.

## Bicoid Homeodomain expression and protein purification

A cDNA coding for amino acids 89–154 of the Bcd protein (including the homeodomain) as described in (*Burz et al., 1998*) using primers

 5'-CAGCCAcatatgCTTTTCGATGAGCGAACG-3'
 5'-GAGCCCggatcccta<u>AGCGTAATCTGGAACATCGTATGGGTA</u>CTGTTTCATACCCGGCGA-3'
 to add a C-terminal HA epitope tag (underlined) and NdeI and BamHI sites (lowercase).

The PCR product was cloned into the pET-15b plasmid using NdeI and BamHI, which contains an N-terminal 6xHis tag and T7 promoter, to make plasmid pET-15B-BcdHD. Expression was induced in BL21 (DE3) pLysS *E. coli* cells using 2 mM IPTG. The protein was purified by affinity chromatography using HisPur Cobalt Resin (Fisher Scientific Cat # 89965) followed by ion exchange chromatography with SP Sepharose Fast Flow resin (GE Healthcare Cat #17-0729-01).

## EMSAs and $K_d$ calculations

EMSAs were performed using purified Bcd homeodomain and biotin-labeled DNA probes were designed to span ~200 bp in the maximal peak region of Bcd-bound peaks identified by ChIP and corresponding to previously characterized enhancers (see *Table 7*). Effective $K_d$ values for each enhancer probe were calculated using the ratio of total shifted probe to free probe.

---

**Table 6.** Estimated nuclear concentrations of Bcd protein in each uniform line.

| Genotype | Protein level relative to WT | Estimated number of molecules | Estimated nuclear concentration |
|---|---|---|---|
| *bcd > uBcd* | 0.14 | 1085 | 12.58 nM |
| *mtrm > uBcd* | 1.1 | 8525 | 98.84 nM |
| *αTub67C > uBcd* | 2.7 | 20925 | 242.61 nM |

DOI: https://doi.org/10.7554/eLife.28275.019

**Table 7.** Primer sequences for EMSA probes.

| Primers | | Sequence (5'- > 3') |
| --- | --- | --- |
| hbP2 probe F | Forward primer | /bio/GTCAAGGGATTAGATGGGCA |
| hbP2 probe R | Reverse primer | /bio/GTCGACTCCTGACCAACGTA |
| kni post F | Forward primer | /bio/AGAAAAAATGAGAACAATGTGAC |
| kni post R | Reverse primer | /bio/AGCCAGCGATTTCGTTACCT |
| kni ant F | Forward primer | /bio/ACAACACCGACCCGTAATCC |
| kni ant R | Reverse primer | /bio/GTCATGTTGGCTAATCTGGC |
| kr ant F | Forward primer | /bio/CAGAAAAGAAAAAGTGTAACGCC |
| Kr ant R | Reverse primer | /bio/GCGAAAAAACGCGTCGCGCT |
| otd intron F | Forward primer | /bio/ATCGTTCCTTGCGGTTTAAT |
| otd intron R | Reverse primer | /bio/AGAACAGGACAAAGGGAATTTAATC |
| otd early F | Forward primer | /bio/CTCGCCTCGCGTGCGACATT |
| otd early R | Reverse primer | /bio/CCTGCGGCAGGACTTCACTT |
| btd F | Forward primer | /bio/ACGAAGTCAAAACTTTTCCA |
| btd R | Reverse primer | /bio/AGCTAAGAGATCTCAACCAAC |
| gt −3 F | Forward primer | /bio/TTACAACTGCCCATTCAGGG |
| gt −3 R | Reverse primer | /bio/GAAGGGCTCGGGTTCGG |
| gt −10 F | Forward primer | /bio/AGATCCAGGCGAGCACTTGA |
| gt −10 R | Reverse primer | /bio/TTAAATTAAAATGTCGCAGGAAGGCG |

DOI: https://doi.org/10.7554/eLife.28275.020

## ChIP-seq and data analysis

### Sample collection

*Drosophila* embryos were collected from 0 to 4 hr laying cages, dechorionated in bleach and cross-linked with 180 mL 20% paraformaldehyde in 2 ml PBS + 0.5% Triton X-100 and 6 mL Heptane for 15 min. Crosslinking was quenched with 125 mM Glycine in PBS + 0.5% Triton X-100. Fixed embryos were visually staged and sorted using a dissecting microscope, and all experimental replicates consisted of 200 embryos in nuclear cycle 14. Chromatin immunoprecipitation was performed with an anti-GFP antibody (Millipore) in embryos expressing GFP-tagged Bcd either in a wild-type graded distribution (*eGFP-Bcd;;bcd^{E1}* [n = 8 biological replicates], *eGFP-Bcd;;bcd^{E1} nos^{L7} tsl^{4}* [n = 2 biological replicates] and *eGFP-Bcd^{085};bcd^{E1}* [n = 3 biological replicates]) or uniformly (*GFP-uBcd;;bcd^{E1}* [n = 3 biological replicates for each uniform line]). Sequencing libraries were prepared using the NEBNext ChIP-seq Library Prep master mix kit and sequenced as described in (*Blythe and Wieschaus, 2015*; *Drocco et al., 2011*).

### Defining a peak list

Barcode split sequencing files were mapped to *Drosophila melanogaster* genome assembly BDGP R5/dm3 using Bowtie2 (*Gregor et al., 2007a*; *Langmead and Salzberg, 2012*) using default parameters. To generate a conservative, high-confidence list of Bcd-bound peaks, peaks were called on each replicate of wild-type and uniform Bcd ChIP-seq data using MACS2 (*Gregor et al., 2007b*; *Zhang et al., 2008*) with settings -p 1e-3 –to-large –nomodel –shiftsize 130 for wild-type samples and -p 0.000001 –slocal 5000 –llocal 50000 –keep-dup all for uBcd samples. The most reproducible peaks from each genotype were selected using an irreproducible discovery rate (IDR) of 1% (*Dubuis et al., 2013*; *Landt et al., 2012*; *Li et al., 2011*). Given evidence that highly transcribed (i. e., highly accessible) regions often give false positive results in ChIP experiments (*Burz et al., 1998*; *Teytelman et al., 2013*), we used our ATAC-seq data to filter our ChIP-seq peaks. We compared the number of CPM-normalized ATAC-seq reads to ChIP-seq reads in each peak, and performed permutation tests (n = 1,000) to determine the probability of selecting open regions of the genome at random that had higher ATAC-seq counts (i.e., regions that were more accessible) than the

ATAC-seq counts in the Bcd ChIP peaks. We determined that at a ratio of 5.4 ATAC-seq/ChIP-seq counts, 95% of the ChIP peaks (permutation test p value = 0.05) were no more open than a random selection of open regions. We filtered out the remaining ChIP peaks with ATAC/ChIP ratios above 5.4, as these peaks are more likely to correspond to highly transcribed open regions where most false positive signals can be found. We then chose the peaks that were common to wild-type and uniform Bcd embryos, which resulted in a list of 1,027 Bcd-bound peak regions. The number of peaks at each step of this filtering is shown in *Table 1*.

## Comparing binding between uniform bcd levels

Mapped BAM files were imported into R as GenomicRanges objects (*Lawrence et al., 2013*), filtering out reads with map quality scores below 30. Significant differences between the uBcd levels were assessed on a pairwise basis using edgeR (*Robinson et al., 2010*) in the 1027 pre-defined peak plus 50,000 additional non-peak noise regions selected from the dm3 genome.

## Data normalization and display

Sequencing data was z-score normalized for display in heatmaps. Sequencing read count coverage was calculated for 10 base pair windows across the genome, and the mean counts per million reads were determined in each ChIP peak, as well as the additional noise peaks. Z-scores were computed for each peak using

$$z = \frac{CPM - \mu}{\sigma}$$

where $\mu$ = mean CPM in noise peaks and $\sigma$ = standard deviation of CPM in noise peaks.

## ATAC-seq and data analysis

### Sample collection

Live embryos expressing a histone (H2Av)-GFP or RFP construct were individually staged on an epifluorescence microscope in halocarbon oil. After the onset of nuclear cycle 14, single embryos were dechorionated in bleach and macerated in cold lysis buffer at t = 12 min into NC14. Samples were pelleted in lysis buffer at 4°C (3000 rpm for 10 min), buffer was removed, and the embryo pellet was flash frozen on dry ice. Frozen pellets were resuspended in Nextera Tagment DNA Buffer +Enzyme and incubated at 37°C for 30 min shaking at 800 rpm. Tagged DNA was purified using a Qiagen Minelute column and eluted in 10 μL. Barcoded sequencing libraries were generated by PCR amplifying the purified DNA using the Nextera DNA Sample Prep kit. Paired-end sequencing was performed on six samples per genotype (n = 5 samples for $bcd^{E1}$) by the Lewis Sigler Institute for Integrative Genomics Sequencing Core Facility on an Illumina HiSeq 2500.

### Data processing

Initial processing of the data was performed essentially as described in (*Blythe and Wieschaus, 2016*). Sequencing files were barcode split and adaptors were trimmed using TrimGalore. Trimmed reads were mapped to the BDGP R5/dm3 genome assembly using BWA (*Li and Durbin, 2009*) with default parameters. Optical and PCR duplicates were marked using Picard Tools MarkDuplicates (https://broadinstitute.github.io/picard/). Mapped reads were filtered using samtools (*Li et al., 2009*) to remove reads with a map quality score ≤30, unmapped reads, improperly paired reads, and duplicate reads. To distinguish reads corresponding to open chromatin reads from nucleosome protected reads, the size of the ATAC-seq fragments were fit to the sum of an exponential and Gaussian distribution as described in (*Buenrostro et al., 2013*). We used a fragment size cutoff of ≤100 bp to identify fragments originating from open chromatin. Filtered open reads were imported into R as GenomicRanges objects.

### Peak calling

Regions of open chromatin at NC14 +12 min were determined by calling peaks on the merged open chromatin reads from wild-type replicates using Zinba (*Rashid et al., 2011*) with the parameters:

 input = 'none', winSize = 300, offset = 50, extension = 65, selectmodel = FALSE, formula = exp_count ~ exp_cnvwin_log + align_perc, formulaE = exp_count ~ exp_cnvwin_log + align_perc,

formulaZ = exp_count ~ align_perc, FDR = TRUE, threshold = 0.05, winGap = 0, cnvWinSize = 2.5E + 4, refinepeaks = TRUE.

## Nucleosome positioning

Nucleosome positioning was determined in samples from all genotypes using NucleoATAC (*Schep et al., 2015*), with default settings. Peak regions used for NucleoATAC were open chromatin peaks from Zinba combined with the Bcd ChIP peaks and widened to 2500 bp centered over the peak maxima. This combined peak list was then reduced and used as NucleoATAC input. For each genotype, BAM files from ATAC-seq were merged and used as input for NucleoATAC.

## Differential Accessibility between Wild-Type, bcd$^{E1}$, and zld$^-$ Embryos

EdgeR was used to determine significant differences in accessibility between different genotypes, with an exact test FDR $\leq$ 0.05 used as the significance cutoff. For edgeR comparisons, the ATAC-seq peaks called by Zinba from wild-type embryos at NC14 +12 min were combined with Bcd ChIP-seq peaks resized to 300 bp centered around the peak summit and 25,000 background regions. The background regions were generated by extending each of the ATAC open peaks to 10 kb and subtracting them from the dm3 genome assembly. The remaining non-peak regions were then sampled randomly 25,000 times and widened to reflect the distribution of sizes in the ATAC-seq peaks. Bcd- or Zld-dependent peaks were those peaks identified as having reduced accessibility in *bcd$^{E1}$* or *zld$^-$* embyos. A summary of the differential accessibility in the ATAC-seq vs. Bcd ChIP-seq peaks is shown in *Table 4*.

To measure differential accessibility of Bcd motifs between wild-type and *bcd$^{E1}$* mutant embryos, the positions of Bcd motifs within ChIP-seq peaks were found, and ATAC-seq accessibility scores were calculated for the 10 bp window containing the midpoint of each Bcd motif. Scores for all motifs within a single peak were averaged prior to plotting. Motifs were found using the 'strong' Bcd position weight matrix via the R function 'matchPWM' in the Biostrings package with an 80% match threshold.

To estimate the overlap between nucleosomes and Bcd motifs, predicted nucleosome dyad centers from NucleoATAC were widened to 160 bp and motifs overlapping these intervals were scored as 'nucleosome associated'. Motifs not overlapping widened nucleosome intervals were scored as 'open'. The fraction of open Bcd motifs per peak was calculated by dividing the number of open Bcd motifs by the total number of encoded Bcd motifs over each peak.

## Data normalization and display

For each genotype, BAM-formatted ATAC-seq reads for each replicate, filtered by quality and duplicates removed as described above, were merged into a single file. Coverage was calculated from these BAM files in 10 bp windows in the dm3 genome assembly. This coverage was then normalized by counts per million reads. The genome coverage in 2 kb regions flanking Bcd ChIP peaks was then z-score normalized as described above. The z-score values are displayed as heatmaps of open chromatin displayed in *Figure 3A and B*. Heatmaps are plotted in an order of decreasing accessibility z score in wild-type embryos.

Predicted dyad centers from NucleoATAC were widened to 147 bp to model nucleosome positions. Occupancy scores from NulcleoATAC in the 147 bp nucleosomes were computed in 10 bp windows across the dm3 genome assembly. Occupancy scores overlapping Bcd ChIP peaks are plotted in the heatmaps in *Figure 4A and B*. Nucleosome heatmaps are plotted in the same order as open chromatin heatmaps, by decreasing accessibility z score in wild-type embryos.

*Supplementary file 1* shows the genomic (BDGP Release 5/dm3) coordinates of Bcd-bound peaks identified by ChIP-seq as described in *Table 1*. Additional columns indicate the nearest gene to the peak that shows maternal or zygotic expression in the embryo (*Blythe and Wieschaus, 2015*) and classifications of each peak as Bcd or Zld dependent for accessibility (determined by ATAC-seq) and their sensitivity group classifications (determined by ChIP-seq). The tileID column gives the name of each Vienna Tile-GAL4 construct (if any) that overlaps with each peak, and the HC_ID column indicates reporter constructs from (*Chen et al., 2012*) that overlap with each peak.

## In situ hybridizations and Cad-GAL4 reporters

The GAL4 coding sequence was amplified from a genomic DNA preparation generated from a *Drosophila* stock carrying a GAL4 reporter, using primers 5'- TGCGATATTTGCCGACTTA-3' and 5'-TGTAATACGACTCACTATAGGGAACATCCCTGTAGTGATTCCA-3'. The amplified sequence was used as a template in the MEGAscript T7 Transcription Kit (ThermoFisher Cat. #AM1334) to with digoxygenin-labeled UTP to generate an anti-GAL4 RNA probe. In situ hybridizations were performed according to standard protocols.

## Accession numbers

The Gene Expression Omnibus (GEO) accession number for this study is GSE86966.

## Acknowledgements

We thank T Gregor, E Gavis, O Grimm, and members of the Wieschaus and Schüpbach labs for helpful feedback and discussion, H Chen, M Levine and S Little for comments on the manuscript, and W Wang and the staff of the Lewis-Sigler Institute Sequencing Core Facility. This work was supported by Ruth L. Kirschstein NRSA pre-doctoral fellowship F31HD082940 (CEH) and postdoctoral fellowship F32HD072653 (SAB).

## Additional information

### Funding

| Funder | Grant reference number | Author |
| --- | --- | --- |
| Howard Hughes Medical Institute | | Eric F Wieschaus |
| National Institutes of Health | F31HD082940 | Colleen E Hannon |
| National Institutes of Health | F32HD072653 | Shelby A Blythe |

The funders had no role in study design, data collection and interpretation, or the decision to submit the work for publication.

### Author contributions

Colleen E Hannon, Conceptualization, Data curation, Formal analysis, Funding acquisition, Investigation, Visualization, Methodology, Writing—original draft, Writing—review and editing; Shelby A Blythe, Conceptualization, Formal analysis, Funding acquisition, Investigation, Visualization, Methodology, Writing—review and editing; Eric F Wieschaus, Conceptualization, Supervision, Funding acquisition, Writing—review and editing

### Author ORCIDs

Colleen E Hannon (iD) http://orcid.org/0000-0002-4402-8107
Eric F Wieschaus (iD) http://orcid.org/0000-0002-0727-3349

### Decision letter and Author response

Decision letter https://doi.org/10.7554/eLife.28275.031
Author response https://doi.org/10.7554/eLife.28275.032

## Additional files

### Supplementary files

• Supplementary file 1: Bicoid ChIP-seq Peak Regions and Functional Annotation The 1027 peak regions as determined by ChIP-seq for Bcd are listed with functional categorizations described in the text. Column names are indicated in the first row. Genomic coordinates of peak binding regions were determined against the dm3 build of the *Drosophila melanogaster* genome, and are reported in columns 1–4 (chromosome, start, end, strand). The nearest zygotically active gene (including

maternal and zygotic genes) is listed in column 5 (nearest.mz). Columns 6 and 7 indicate whether a Bcd ChIP-seq peak was determined to require either Bcd (column 6) or Zld (column 7) for chromatin accessibility as determined by ATAC seq. Columns 8–11 indicate the concentration-dependent binding class determined for each peak. Column 12 indicates the identification number of Vienna Tiles that were found to uniquely overlap with a Bcd ChIP-seq peak. Column 13 indicates the identification number of DNA fragments reported in *Chen et al. (2012)* overlapping with Bcd ChIP-seq peaks from this study. Column 14 indicates the Bcd ChIP-seq peaks that were found to overlap with Zld ChIP-seq peaks reported in *Harrison et al. (2011)*.

DOI: https://doi.org/10.7554/eLife.28275.021

• Transparent reporting form
DOI: https://doi.org/10.7554/eLife.28275.022

### Major datasets
The following dataset was generated:

| Author(s) | Year | Dataset title | Dataset URL | Database, license, and accessibility information |
|---|---|---|---|---|
| Hannon CE, Blythe SA, Wieschaus EF | 2016 | Concentration dependent binding states of the Bicoid Homeodomain Protein | https://www.ncbi.nlm.nih.gov/geo/query/acc.cgi?acc=GSE86966 | Publicly available at the NCBI Gene Expression Omnibus (accession no. GSE86966) |

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
