## [Decision Letter]

Thank you for submitting your article "Concentration Dependent Chromatin States Induced by the Bicoid Morphogen Gradient" for consideration by *eLife*. Your article has been favorably evaluated by James Manley (Senior Editor) and three reviewers, one of whom is a member of our Board of Reviewing Editors.

The reviewers have discussed the reviews with one another and the Reviewing Editor has drafted this decision to help you prepare a revised submission.

The manuscript shows a strong dataset demonstrating that Bicoid influences chromatin accessibility in a concentration-dependent manner, and that Bcd can act as a pioneer factor when at high concentrations. Using fly embryos engineered to uniformly express high, medium, and low levels of Bcd protein, the authors identified four classes of Bcd bound regions, with Concentration-Sensitive I and II classes bound only by high and high/medium levels of Bcd, respectively, driving anterior expression of candidate enhancer fragments, enriched for Bcd motifs, and highly represented in Bcd-dependent chromatin accessibility regions. Concentration-Sensitive III class peaks were bound stronger by high/medium levels of Bcd but lower by low levels of Bcd, driving broad expression of candidate enhancer fragments, not enriched for the transcription factor Zelda (Zld) or Bcd motifs, and not quite dependent on Bcd for chromatin accessibility. Concentration-Insensitive class peaks were bound equally well by all levels of Bcd, driving posterior expression of candidate enhancer fragments, highly enriched for Zld motifs, and not dependent on Bcd for chromatin accessibility. The authors found that the presence and/or the strength of Bcd motifs do not account for in vivo sensitivity to Bcd concentration, as predicted by the binding site affinity threshold model for morphogen function. Rather, the Bcd morphogen gradient is read at the level of chromatin accessibility, established at Bcd concentration-sensitive targets primarily and perhaps even independently by high levels of Bcd, and at concentration-insensitive targets by factors like Zld. The authors further demonstrated using a C-terminus truncated Bcd that the ability of high levels of Bcd to drive chromatin accessibility at concentration-sensitive targets is likely through co-factor recruitment, rather than Bcd cooperative binding. Finally, they turn a Zld dependent target enhancer into a Bcd high level target by changing Zld binding sites into Bcd binding sites thereby increasing the number of Bcd binding sites, and suggest that high levels of Bcd can confer chromatin accessibility.

Overall, the paper supports the model put forth, whereby Bcd itself, acting through different classes of genomic binding sites differentially sensitive of Bcd concentration, can create a regulatory logic that delivers spatio-specific regulation of its targets.

Although the study was well designed and performed, the reviewers ask that the authors address several criticisms as outlined below.

1) The manuscript was difficult to follow at times. For Bcd aficionados it may be easier, but for the more general audience of *eLife*, it might be difficult to read, not only because of the many typos, but the jargon and flow. Examples are: a) What does "strong Bcd gradient" mean? b) Introduction, third paragraph, first sentence: The two phrases seem disconnected. Also, the last sentence of this paragraph could also use some word changes to be more logical; perhaps "Therefore" instead of "However" would flow better. c) A transition is missing between the last two paragraphs of the Introduction. Why is the employed approach better than computational predictions and in vitro measurements? The logic is not obvious. d) Introduction, last paragraph: The use of the word "enhancer" is vague and somewhat misleading. The data reveal distinct classes of Bcd targets not necessarily enhancers. However, further down when referring to the model, it seems appropriate to use "target enhancers."

2) First section in Results is titled: "Bicoid target gene…other maternal factors, but is physical interaction is not”. Change "maternal" to "patterning" since Bcd binding is changed in zelda mutants (Xu et al., 2014).

3) With regard to "enhancers whose expression patterns span broadly across the AP axis" – where is the data to support this statement. How many of the 1027 Bcd peaks do these enhancers you speak of comprise – 10 or 100? It is unclear if the following section, are the three specific targets supposed to represent these enhancers? It is not clear because the transition is poor between these paragraphs. Nevertheless, please show the data that led to this conclusion that peaks "overall associate with enhancers[…]"

4) The terminology used in the Results (and elsewhere) is incorrect. "the *kni* posterior enhancer is not expressed" – enhancers are not expressed, genes are expressed; enhancers drive expression. By the way, which enhancer is driving the strong expression seen in the *bcd hb nos tsl* embryo in Figure 1 bottom right? Why is there such a discrepancy between this embryo and the one shown in Figure 1—figure supplement 1 where expression appears lower? Please sort this out.

5) In general, the term "enhancer" is used too loosely in this paper and it is advised that the authors change this. For example, in the last sentence of the second paragraph of the subsection “Bcd binding to genomic targets is concentration dependent” the use of "enhancer" (twice) is premature because it isn't until the following paragraph where a correlation between Bcd peaks and known enhancers is discussed.

6) In the third paragraph of the subsection “Bcd binding to genomic targets is concentration dependent”, of the 163 Vienna enhancers that are active in stage 4-6, how many are expressed in a discrete AP domain? I could not find this information and it should be included. Also, why don't all 163 candidate enhancers overlap with a Bcd peak if the 293 overlap with at least one Bcd peak. Also, it is unclear how "77.2%" was derived. 163/293 = 55.6. Please clarify.

7) The statement, "we find no evidence that the Bcd sensitivity classes are predominantly defined by repressive interactions." It is worrysome that the authors are making a sweeping statement when there are clearly Concentration-Sensitive I enhancers that bind repressors, in addition to those shown by Chen et al. (2012). For example, anterior *Gt* and *Kr* are not in the same expression domain, which is not in align with the statement in the last paragraph of the subsection “Bcd binding to genomic targets is concentration dependent”, and *Kr* was found enriched in Concentration-Sensitive I peaks that include the *Gt* anterior enhancer. Though it is brave to conclude from the data presented that Bcd sensitivity class gene expression domains are not defined by repressors, it seems dangerous because so many AP gene domains that have been well studied are defined by repressor interactions. Is it possible that enrichment for specific repressors was not seen because your group of peaks contains peaks that are not real Bcd target enhancers and thus the real ones get diluted out?

8) In Figure 2—figure supplement 2 and B that Concentration-Sensitive III peaks are enriched in pol II binding and the Dref motif, which is a type of promoter element. Also, it is not clear why the BDTNP Bcd peaks were not enriched in Concentration-Sensitive III peaks even though they are in this study. I'm somewhat confused by these peaks – on the one had they appear to be at promoters (versus enhancers), which might bind Bcd promiscuously because they are so open, but on the other hand they have Bcd binding sites. At any rate, it would be important to show more examples of the genes associated with Concentration-Sensitive III peaks and what happens to their expression in bcd mutants, starting with the two mentioned, *cnc* and *hkb*. Also, importantly, it should be shown how many of the peaks in each class are near a TSS (within 500 bp). Do they differ?

9) Text in the first paragraph of the subsection “Sequence composition of ChIP sensitivity classes does not account for in vivo sensitivity to Bcd concentration” is confusing. The authors say their results are not in agreement with those of Ochoa-Espinosa et al. (2005) study, but then in the next paragraph they say their results confirm theirs. Please rewrite for clarity.

10) When demonstrating the relationship between Zld and Bcd at Bcd-bound regions, the authors claimed that "Zld-dependent peaks are distributed across each sensitivity class determined by ChIP, with no particular class being significantly more Zld dependent," thus Zld "is unlikely to determine the differential concentration sensitivity of Bcd peaks as a whole." This should be examined more closely. How many of the 855 Bcd peaks that overlap with ATAC open regions overlap with Zld binding? How many of the 402 Bcd peaks with reduced accessibility in *zld-* overlap with Zld binding? How many ATAC-seq peaks overlap with Zld binding? The authors are advised to show Zld binding, Zld motifs, and Bcd motifs in Figure 3, and/or provide additional analysis separating the four classes into those bound by Zld and not bound by/bound by low levels of Zld. Does the conclusion change, e.g., is any Bcd peak class significantly more Zld dependent? If so, this must be added.

11) Structure function studies (protein deletion mutant here) are often difficult to assess because it is unclear whether the structure, and thus function, of the remaining protein (in this case the homeodomain), not just its presence, is unchanged, which is beyond the scope of this experiment. Therefore, the authors should qualify their conclusion by adding something like "assuming the structure of the homeodomain remains completely intact…"

12) The authors showed that a Concentration-Insensitive target caudal (cad), whose chromatin accessibility is Zld-dependent and Bcd-independent, can be converted to a Bcd concentration-sensitive target at anterior region by mutating Zld motifs into Bcd motifs. The authors reasoned that high Bcd levels in anterior regions conferred chromatin accessibility to the transgene. If Bcd-dependent chromatin accessibility at regions with high levels of Bcd is the key to its target expression, the authors should provide additional data to show that Bcd binds stronger to Cad (Zld->Bcd)-GAL4 than to Cad-GAL4 in both wt and *zld-* embryos; 2) that accessibility increases when adding Bcd sites (in the absence of Zld sites); and 3) though elaborate, adding Bcd sites to HC-45 instead of Zld sites should render the enhancer active if Bcd can pioneer chromatin.

13) Discussion of model in Figure 7. The idea that in regions where morphogen levels are low, Zld is required to boost activity has been proposed for both Dl and Bcd (Nien et al., 2011). Likewise, that morphogens at high levels can access and activate their high-level targets mostly on their own has also been discussed (same paper). This study corroborates and advances those ideas very nicely, and this should be mentioned.

14) The authors should move the last sentence of the paper “This may be a common property of developmental transcription factors that must gain early access *to* their target enhancers while the chromatin state of the genome is being remodeled during large scale transitions in the gene regulatory landscape.” to the Abstract.

[Editors' note: further revisions were requested prior to acceptance, as described below.]

Thank you for resubmitting your work entitled "Concentration Dependent Chromatin States Induced by the Bicoid Morphogen Gradient" for further consideration at *eLife*. Your revised article has been favorably evaluated by James Manley (Senior Editor), a Reviewing Editor, and 1 reviewer.

The manuscript has been improved but there are some remaining issues that need to be addressed before acceptance, as outlined below:

The authors have satisfactorily addressed reviewer concerns. The efforts made to address further experiments (especially those suggested for the cad reporter) were greatly appreciated.

Answers to author questions are answered below:

1) Regarding the Kni stainings: We prefer to use the Kni stainings from the first experiment in Figure 1, because that experiment also utilized the same patterning markers (e. g., Btd) used in the other genotypes of the figure. We could move the Supplementary experiment to the main Figure 1, but would prefer to keep it in the Supplement. We have added sentences to the text and to the figure legend to point the reader to the more supplementary material.

Reviewer's answer: Yes, please keep it in the supplement.

2) Regarding the repressor binding: However, we agree that the data are not entirely clear, and difficult to interpret. Overall we agree that this analysis does not advance the concepts in the manuscript. We did not make it a main figure in the paper for this reason, and one possibility would be to remove the figure and corresponding argument entirely from the last paragraph of the subsection “Bcd binding to genomic targets is concentration dependent”. We would appreciate advice from the reviewers on this point.

Reviewer's answer: Please delete the corresponding argument from the last paragraph of the subsection “Bcd binding to genomic targets is concentration dependent”, rather than keeping the new paragraph because it isn't much better with the single repressor focus.

3) Regarding the assays on cad and HC45 reporters: The major limitation to measuring chromatin accessibility or performing ChIP on reporter constructs (as suggested in experiments 1 and 2 for our cad reporters) is that the measurements must be able to distinguish between the transgenic (reporter) and endogenous (wild-type) copies of the regulatory element in question.

Reviewer's answer: While it is possible to distinguish transgenic from endogenous enhancers by having one primer in the reporter sequence (GAL4), which is likely within ChIPed fragments, and one inside the enhancer (as in Xu et al., 2014 for Bcd ChIP on HC45), it is not necessary pursuing this venue at this point.

With regard to the new HC45 data, the discussion about endogenous HC45 being accessible whether active or not and with or without Zld or Bcd is interesting, though it is formerly possible that local chromatin at the transgene is not behaving the same way as endogenous. Reviewers recommend having this data as a supplemental figure.

---

## [Author Response]

[…] Although the study was well designed and performed, the reviewers ask that the authors address several criticisms as outlined below.1) The manuscript was difficult to follow at times. For Bcd aficionados it may be easier, but for the more general audience of eLife, it might be difficult to read, not only because of the many typos, but the jargon and flow. Examples are: a) What does "strong Bcd gradient" mean? b) Introduction, third paragraph, first sentence: The two phrases seem disconnected. Also, the last sentence of this paragraph could also use some word changes to be more logical; perhaps "Therefore" instead of "However" would flow better. c) A transition is missing between the last two paragraphs of the Introduction. Why is the employed approach better than computational predictions and in vitro measurements? The logic is not obvious. d) Introduction, last paragraph: The use of the word "enhancer" is vague and somewhat misleading. The data reveal distinct classes of Bcd targets not necessarily enhancers. However, further down when referring to the model, it seems appropriate to use "target enhancers."

We have made changes throughout the revised manuscript (including those specifically listed in the point above) to make the manuscript more readable and easier to follow.

2) First section in Results is titled: "Bicoid target gene…other maternal factors, but is physical interaction is not”. Change "maternal" to "patterning" since Bcd binding is changed in zelda mutants (Xu et al., 2014).

This wording has been changed to follow the suggestions of the reviewers.

3) With regard to "enhancers whose expression patterns span broadly across the AP axis" – where is the data to support this statement. How many of the 1027 Bcd peaks do these enhancers you speak of comprise – 10 or 100? It is unclear if the following section, are the three specific targets supposed to represent these enhancers? It is not clear because the transition is poor between these paragraphs. Nevertheless, please show the data that led to this conclusion that peaks "overall associate with enhancers…"

The description of expression patterns that span broadly across the AP axis was meant to refer to the 66 previously identified Bcd-dependent enhancers that overlap with our ChIP-seq peaks. However, upon re-reading this was understandably confusing. We have re-written this section for clarity.

4) The terminology used in the Results (and elsewhere) is incorrect. "the kni posterior enhancer is not expressed" – enhancers are not expressed, genes are expressed; enhancers drive expression. By the way, which enhancer is driving the strong expression seen in the bcd hb nos tsl embryo in Figure 1 bottom right? Why is there such a discrepancy between this embryo and the one shown in Figure 1—figure supplement 1 where expression appears lower? Please sort this out.

The embryos of different genotypes in Figure 1 are from different stainings, and compiled over a ~six month period as the genotypes became available. Although stained with identical protocols and antibodies, they were intended to compare spatial patterns of expression. They are not directly quantitatively comparable. In Author response image 1 we have included images of four *bcd hb nos tsl* embryos stained for Kni from the experiment shown in Figure 1, with the embryo used in the figure outlined in red. We have also included a wild-type embryo stained in parallel (though in a separate tube).

We attempted to choose a representative image for this genotype in Figure 1. However, as is evident here, the *kni* expression in this experiment is variable and the levels in some cases were higher than the background levels in the separately stained wild-type controls. This is perhaps due to the small number of germ line clone derived embryos *bcd hb nos tsl* stained in the tube.

To address this issue, we therefore repeated this experiment to allow for more quantitative comparisons of *kni* expression, simultaneously staining all genotypes (wild-type, *hb nos tsl*, and *bcd hb nos tsl*) in the same experiment, mixed in a single tube. In the original manuscript, we presented this data as a supplement to Figure 1, and quantified with a quantification of *kni* expression along the AP axis. These data clearly show that *kni* expression is reduced to low levels in the absence of Bcd, supporting the possibility that Bcd binding to the posterior *kni* enhancer is essential for the expression it drives.

We prefer to use the Kni stainings from the first experiment in Figure 1, because that experiment also utilized the same patterning markers (e.g., Btd) used in the other genotypes of the figure. We could move the supplementary experiment to the main Figure 1, but would prefer to keep it in the supplement. We have added sentences to the text and to the figure legend to point the reader to the more supplementary material.

5) In general, the term "enhancer" is used too loosely in this paper and it is advised that the authors change this. For example, in the last sentence of the second paragraph of the subsection “Bcd binding to genomic targets is concentration dependent” the use of "enhancer" (twice) is premature because it isn't until the following paragraph where a correlation between Bcd peaks and known enhancers is discussed.

We have changed our use of "enhancer" throughout the revised manuscript in instances where it was used prematurely or inappropriately.

6) In the third paragraph of the subsection “Bcd binding to genomic targets is concentration dependent”, of the 163 Vienna enhancers that are active in stage 4-6, how many are expressed in a discrete AP domain? I could not find this information and it should be included. Also, why don't all 163 candidate enhancers overlap with a Bcd peak if the 293 overlap with at least one Bcd peak. Also, it is unclear how "77.2%" was derived. 163/293 = 55.6. Please clarify.

All 163 candidate enhancers do overlap with Bicoid peaks. But in some cases, more than one Vienna Tile overlaps with a single peak, and in other cases a single Tile overlaps with more than one Bcd peak. The correlation is not one-to-one, so the 163 enhancers only overlap with 151 total Bcd peaks, even though they all overlap with at least one.

Author response image 2 is a screenshot from the UCSC genome browser near *gt*, illustrating the issue of multiple overlaps. Here we see that while VT55790 and VT55792 each overlap a single Bcd peak, that same Bcd peak overlaps with two tiles. The Bcd peak furthest from *gt*, however, overlaps with only one tile, VT55795. VT55790 and VT55792 were therefore excluded from the plot in Figure 2, as the expression patterns that they drive cannot be connected unambiguously to a single Bcd peak.

**Author response image 2. respfig2:** 

The differences in Bcd peak number and Vienna Tile number are shown in Table 2, which is now referred to in the text of the manuscript and should clarify this issue. This table has also been expanded to include the number of single overlaps. Additionally, we returned to the Fly Enhancer database and visually scored the number of Tiles that drive expression in an AP pattern in blastoderm embryos. This has also been added to Table 2.

The correct number of Bcd peaks that do not correspond to any enhancer candidates in the Fly Enhancer collection is 793 (1,027-234), or 77.2%. The reference to 876 non-overlapping in the text is the sum of the peaks not represented as Tiles [793] + peaks overlapping with Tiles active later in development [42] + the peaks overlapping with inactive Tiles [41]. We thank the reviewers for pointing out the confusion that allowed us to detect and correct this error has been corrected in the manuscript.

7) The statement, "we find no evidence that the Bcd sensitivity classes are predominantly defined by repressive interactions." It is worrysome that the authors are making a sweeping statement when there are clearly Concentration-Sensitive I enhancers that bind repressors, in addition to those shown by Chen et al. (2012). For example, anterior Gt and Kr are not in the same expression domain, which is not in align with the statement in the last paragraph of the subsection “Bcd binding to genomic targets is concentration dependent”, and Kr was found enriched in Concentration-Sensitive I peaks that include the Gt anterior enhancer. Though it is brave to conclude from the data presented that Bcd sensitivity class gene expression domains are not defined by repressors, it seems dangerous because so many AP gene domains that have been well studied are defined by repressor interactions. Is it possible that enrichment for specific repressors was not seen because your group of peaks contains peaks that are not real Bcd target enhancers and thus the real ones get diluted out?

We included this analysis in an attempt to address the Chen 2012 model that repressors define the expression domains of Bcd target genes. The data in Figure 2—figure supplement 1 suggest that while repressors interact with Bcd targets in each class, it is not clear from the binding data that these repressors determine the concentration sensitivity of the Bcd ChIP classes. If, for example, Runt were responsible for setting the posterior boundaries of anterior Bcd targets, we would expect enrichment for Runt binding in the Concentration-Sensitive I and II classes, which we do not observe. That was the point we were trying to make. However, we agree that the data are not entirely clear, and difficult to interpret. Overall we agree that this analysis does not advance the concepts in the manuscript. We did not make it a main figure in the paper for this reason, and one possibility would be to remove the figure and corresponding argument entirely from the revised manuscript. We would delete this text in the last paragraph of the subsection “Bcd binding to genomic targets is concentration dependent”. We would appreciate advice from the reviewers on this point.

8) In Figure 2—figure supplement 2 and B that Concentration-Sensitive III peaks are enriched in pol II binding and the Dref motif, which is a type of promoter element. Also, it is not clear why the BDTNP Bcd peaks were not enriched in Concentration-Sensitive III peaks even though they are in this study. I'm somewhat confused by these peaks – on the one had they appear to be at promoters (versus enhancers), which might bind Bcd promiscuously because they are so open, but on the other hand they have Bcd binding sites. At any rate, it would be important to show more examples of the genes associated with Concentration-Sensitive III peaks and what happens to their expression in bcd mutants, starting with the two mentioned, cnc and hkb. Also, importantly, it should be shown how many of the peaks in each class are near a TSS (within 500 bp). Do they differ?

There are several possible reasons why our Bcd ChIP peak list differs from the BDTNP peaks.

First, in the manuscript, we compared our ChIP-seq data to BDTNP ChIP-chip data. Second, the BDTNP data were collected from roughly staged embryos (2-3 hour staging), using two replicates per antibody (MacArthur et al., 2009). Our embryos are more carefully staged and limited to early NC14. Third, we used different criteria to select our peaks. After performing the ChIP-seq experiments, we selected the most reproducible peaks across 8 replicates to generate the peak list. We selected these peaks using the Irreproducible Discovery Rate (IDR) approach used by modENCODE to measure reproducibility between genomic datasets, as detailed in the Materials and methods. We are therefore using a different significance threshold to determine which peaks we include in our list, and this approach is different than the FDR ranking used for the BDTNP ChIP data (MacArthur et al., 2009). As such, we expect to generate a list of peaks that differs from theirs. However, as all of the peaks included in our list are highly reproducible across replicates, we are confident that our dataset allows accurate identification of real Bcd peaks at this stage of development. Overall, the Concentration-Sensitive I and II classes had higher ChIP signals than the Concentration-Sensitive III and Insensitive peaks, so these peaks were less likely to be detected in the BDTNP data. Although we acknowledge that some proportion of these types of peaks could be artifacts due to high accessibility, we have filtered our peaks by ATAC-seq accessibility score to mitigate this issue (described in the Materials and methods).

With respect to the proximity of these peaks to promoters, many early developmental genes have cis-regulatory elements in close proximity to their promoters, for example hunchback P2. It is therefore not clear whether proximity to a promoter rules out a peak as an enhancer, or predicts whether TF binding will be functional. We have added an additional table to the manuscript indicating the number peaks in each class overlapping with Vienna Tile-GAL4 reporters active at stage 4-6 and within 500 bp of a TSS. In both cases there are interesting differences between the sensitivity classes, and these differences are now discussed in the text.

Gene expression patterns for the Concentration Sensitive III members *hkb* and *cnc* have been previously published and shown to be sensitive to Bcd. In *bcd^E1^* mutants, expression of *cnc* is lost (Löhr et al., 2009), while anterior *hobo* expression is confined to a smaller expression domain. (Reuter and Leptin, 1994). We originally intended to include Hkb immunostaining in Figure 1, but had technical difficulties with our antibody.

9) Text in the first paragraph of the subsection “Sequence composition of ChIP sensitivity classes does not account for in vivo sensitivity to Bcd concentration” is confusing. The authors say their results are not in agreement with those of Ochoa-Espinosa et al. (2005) study, but then in the next paragraph they say their results confirm theirs. Please rewrite for clarity.

The Ochoa-Espinosa study concluded that neither the *number* nor *strength* of Bcd binding sites in an enhancer correlated with the posterior boundary position of gene expression driven by that enhancer. Here we demonstrate that Concentration-Sensitive I and II targets are both enriched for enhancers driving anterior expression and more strongly enriched for Bcd binding sites. Our study therefore disputes the finding that the *number* of binding sites in enhancers does not correlate with gene expression pattern, but supports the finding that the *strength* of binding sites does not. This section has been re-written to clarify these two separate points.

10) When demonstrating the relationship between Zld and Bcd at Bcd-bound regions, the authors claimed that "Zld-dependent peaks are distributed across each sensitivity class determined by ChIP, with no particular class being significantly more Zld dependent," thus Zld "is unlikely to determine the differential concentration sensitivity of Bcd peaks as a whole." This should be examined more closely. How many of the 855 Bcd peaks that overlap with ATAC open regions overlap with Zld binding? How many of the 402 Bcd peaks with reduced accessibility in zld- overlap with Zld binding? How many ATAC-seq peaks overlap with Zld binding? The authors are advised to show Zld binding, Zld motifs, and Bcd motifs in Figure 3, and/or provide additional analysis separating the four classes into those bound by Zld and not bound by/bound by low levels of Zld. Does the conclusion change, e.g., is any Bcd peak class significantly more Zld dependent? If so, this must be added.

Using a published list of Zld ChIP-seq peaks (Harrison et al., 2011), we find that 732 of our Bcd ChIP peaks (71.3%) overlap with Zld peaks. Interestingly, despite the higher representation of Zld motifs in the less sensitive classes, the peaks in the more sensitive classes (I and II) actually overlap with more Zld ChIP peaks:

Sensitive I = 125 (82.2%) Zld bound

Sensitive II = 113 (81.9%) Zld bound

Sensitive III = 396 (66.8%) Zld bound

Insensitive = 97 (67.8%) Zld bound

However, in terms of Zld dependence, the more relevant feature is whether a peak is open or closed in the absence of Zld. By this metric, none of the peak classes are significantly overrepresented in the Zld dependent group. A barplot showing enrichment of each class in the Zld dependent peaks has been added to Figure 3, alongside the plot of Bcd dependent peaks. We have also added a supplemental figure (Figure 3—figure supplement 1) showing overlap between Zld and Bcd ChIP peaks and comparisons of Bcd and Zld dependence (as defined by ATAC-seq) between Bcd peaks that are bound or not bound by Zld. These comparisons show that while there is a greater enrichment of Zld binding at the Concentration Sensitive I and II classes, only the Concentration Insensitive class shows a (slight) enrichment for Zld dependence when the classes are separated into those bound and not bound by Zld.

Of all ATAC-seq peaks that are open at NC14, 5182 (39.2%) overlap with Zld ChIP peaks.

However, only 2675 (20.2%) are dependent upon Zld for accessibility (closed in *zld^–^* embryos). We see the same pattern with the Bcd ChIP peaks: although 732 (71.3%) of Bcd ChIP peaks overlap with Zld ChIP peaks, only 402 (39.1%) are dependent on Zld for accessibility.

Of the 855 Bcd peaks that overlap with ATAC open peaks, 654 (76.5%) overlap with Zld binding. Of the 402 with reduced accessibility in *zld^–^* embryos, 371 (92.3%) overlap with Zld binding. Therefore, reduced accessibility in the absence of Zld appears to be a strong predictor of Zld binding, while Zld binding itself is not necessarily as predictive of Zld dependence, in agreement with (Schulz et al., 2015).

Zld and Bcd motif representation in each class was included in the original manuscript in Figure 2—figure supplement 2, which shows that Zld is the most highly represented motif in the Concentration-Insensitive and Concentration-Sensitive III classes, while Bcd is the top motif for the Concentration-Sensitive II and I classes.

11) Structure function studies (protein deletion mutant here) are often difficult to assess because it is unclear whether the structure, and thus function, of the remaining protein (in this case the homeodomain), not just its presence, is unchanged, which is beyond the scope of this experiment. Therefore, the authors should qualify their conclusion by adding something like "assuming the structure of the homeodomain remains completely intact…"

We have now qualified this conclusion in the manuscript.

12) The authors showed that a Concentration-Insensitive target caudal (cad), whose chromatin accessibility is Zld-dependent and Bcd-independent, can be converted to a Bcd concentration-sensitive target at anterior region by mutating Zld motifs into Bcd motifs. The authors reasoned that high Bcd levels in anterior regions conferred chromatin accessibility to the transgene. If Bcd-dependent chromatin accessibility at regions with high levels of Bcd is the key to its target expression, the authors should provide additional data to show that Bcd binds stronger to Cad (Zld->Bcd)-GAL4 than to Cad-GAL4 in both wt and zld- embryos; 2) that accessibility increases when adding Bcd sites (in the absence of Zld sites); and 3) though elaborate, adding Bcd sites to HC-45 instead of Zld sites should render the enhancer active if Bcd can pioneer chromatin.

We agree that being able to perform experiments 1 and 2 would be valuable. However, these experiments present significant technical issues, as described in detail below. Experiment 3 was performed, and confirms the hypothesis that addition of Bcd sites to HC45 renders the enhancer “active”. We will discuss experiment 3 first, followed by our attempt at experiments 1 and 2.

To perform experiment 3, we generated three HC45 transgenic reporter lines (wild-type, +4 Zld sites, +4 Bcd sites). We used the HC45 wild-type and +4 Zld site sequences reported in Xu et al., 2014, and introduced 4 Bcd sites at the same positions used to make the +4 Zld site reporter to make the new +4 Bcd site reporter. We replicated the previously reported result, that HC45 is inactive in blastoderm embryos, and that adding 4 Zld sites confers activity to the reporter. Similar to addition of Zld sites, addition of Bcd sites also confers activity to the reporter (see Figure 6—figure supplement 1).

We performed this experiment at the suggestion of the reviewers, and are encouraged by the positive result, but we have some reservations about whether this provides stronger evidence of “pioneering” activity than our prior results with the cad enhancer. HC45, while inactive in blastoderm embryos, has open chromatin at this stage according to our ATAC data. The wildtype sequence has one Zld binding site, and seven Bcd binding sites. HC45 requires neither Zld (as cad does) nor Bcd for chromatin accessibility, in the sense that it retains chromatin accessibility in embryos maternally mutant for either factor. As such, it is unclear what drives the phenomenon of HC45 “acquiring activity” upon addition of Zld or Bcd sites. By analogy to the hsp70 system, where GAGA-factor pioneers the acquisition of open chromatin, thus allowing HSF to interact with binding sites and activate transcription, it would appear that neither Zld nor Bcd are operating like a pioneer (GAGA-factor), and are instead possibly functioning like transcriptional activators (HSF). On the other hand, we cannot rule out that addition of Zld or Bcd binding sites further enhances chromatin accessibility at HC45, or drives remodeling of chromatin to favor an ‘active’ versus an ‘inactive’ state. To address such questions, we would need to be able to measure differences in binding state and chromatin accessibility at these reporters (as suggested in experiments 1 and 2 above), and as described below, this is technically difficult. For these reasons, we have chosen to incorporate this new result with HC45 only as a supplement to Figure 6.

The major limitation to measuring chromatin accessibility or performing ChIP on reporter constructs (as suggested in experiments 1 and 2 for our cad reporters) is that the measurements must be able to distinguish between the transgenic (reporter) and endogenous (wild-type) copies of the regulatory element in question. This effectively limits our analysis to fragments of DNA that we can distinguish by SNPs. To pursue this, we sequenced the cad enhancer in genomic DNA preps from single adult flies, from every possible genetic background that would be present in the flies used in this experiment. The reporter constructs were generated by injecting plasmid DNA into flies with an attp2 landing site and then crossed to white females to isolate transformants. *zld* germline clones were produced by crossing *zld^294^*/FM7 females to C(1)Dx/*ovo^D^* males. By sequencing genomic DNA from each of these genotypes and comparing it to our cad enhancer reporter sequences, we identified only one SNP that was present in the reporters and absent in all other genotypes. We were therefore constrained to using this SNP to design primers for qPCR. We tested these primers on genomic DNA, and the result is shown in Author response image 3.

**Author response image 3. respfig3:** 

Primer set 1 amplifies a product using the "wild-type" or non-transgenic SNP, primer set 2 amplifies a product using the transgenic SNP, and primer set 3 amplifies a product in the cad enhancer that is identical between the wild-type and transgenic sequences. These primers were successful at distinguishing between the wild-type and transgenic SNP sequences in genomic DNA, as primer 2 only gives a product in DNA from a fly with the cad-GAL4 construct. Similar results were obtained by qPCR on intact genomic DNA.

A second technical limitation is how to effectively measure differences in chromatin accessibility at a single locus by a qPCR based assay that can leverage the presence of the SNP to distinguish between the transgenic and endogenous enhancer copies. ATAC-seq sample preparations are themselves not suitable for PCR-based quantification approaches due to the insertion of adapter sequences at the time of chromatin fragmentation at accessible sites. On the other hand, the feasibility of Formaldehyde-Assisted Isolation of Regulatory Elements (FAIRE)-qPCR has to our knowledge not been reported in the literature, but it is likely that samples prepared by the FAIRE-seq technique would remain compatible with qPCR analysis. We therefore pursued this possibility in order to address this question. We fixed and sorted embryos by stage, homogenized, and sonicated genomic DNA as described for ChIP-seq in the Materials and methods. Following sonication, we isolated soluble chromatin by phenol-chloroform extraction as described in (Simon et al., 2013). We tested the effectiveness of this approach using wild-type (Oregon-R) embryos, and qPCR primers that would give predictable results based on chromatin accessibility measured by ATAC-seq in this study. We used three optimized primer sets that amplify short regions around the hunchback coding sequence. These regions amplified by these primers should be highly accessible (hb ChIP2), accessible (hb ChIP2), and inaccessible (hb P2 +4) at nuclear cycle 14. A plot showing the enrichment of the qPCR products over an input chromatin sample is shown in Author response image 4, and gave the expected results.

**Author response image 4. respfig4:** 

This result was encouraging. To perform the requested experiment to measure accessibility at cad (WT) and cad(*zld-*>*bcd*), we performed FAIRE on wild-type (white) embryos and embryos expressing either the cad-GAL4 or cad(*zld->bcd*)-GAL4 in either a wild-type or *zld^–^* background. Unfortunately, when we performed the FAIRE experiment, the primers that effectively distinguish between the transgenic and endogenous cad enhancer in intact genomic DNA do not distinguish between the wild-type and transgenic alleles on a sheared DNA sample. Both the wild-type and transgenic primer sets gave a low signal compared to the highly accessible hb ChIP2 region, and are indistinguishable between embryos with and without the transgenic reporter:

**Author response image 5. respfig5:** 

Since we cannot even detect the transgene (or the endogenous copy) in a sheared DNA prep, and since we are limited to this single priming site to distinguish the alleles of this enhancer, we feel that we have exhausted our options for measuring the accessibility of the transgenic reporters directly, and are technically unable to address this point sufficiently.

Similarly, to measure differential binding of Bicoid to these reporters by ChIP (suggested experiment 1), we would be constrained by the same technical limitations as described for measuring chromatin accessibility via SNP based PCR. We emphasize that we recognize the potential power of such an experimental approach, but regrettably for this particular set of reporter constructs, despite our hardest efforts, we cannot address these questions within the limited time available for revision of this manuscript.

What is the most likely explanation for our results with the cad enhancer? Our data demonstrate that this enhancer is Zld-dependent, meaning that in *zld* mutant embryos, neither is the enhancer accessible, nor is the enhancer active. Suggested experiment 1 is designed to confirm an expected result, namely that adding Bcd sites results in greater Bcd binding in both wild-type and *zld* mutant embryos. This is likely to be the case. Suggested experiment 2 is designed to show the presumed positive correlation between chromatin accessibility and the activity state of the reporter construct. We are not aware of any documented example of an enhancer that shows activity without an accessible chromatin signature. Similar to the wild-type enhancer in *zld* mutant embryos, mutating the Zld motifs to a non-functional DNA sequence eliminates the activity of this enhancer in wild-type embryos (seeFigure 6—figure supplement 1A). When we instead convert the Zld motifs to Bcd motifs (Z->B), we recover enhancer activity, albeit in a different spatial pattern (Figure 6). The (Z->B) enhancer is active in *zld* mutants, and is now completely dependent on Bcd for activity (Figure 6). In sum, we feel that it is highly likely that the recovery of activity of the Cad(Z->B) enhancer stems from Bcd-dependent acquisition of chromatin accessibility, and that the shift in the spatial pattern reflects the demonstrated concentration-dependence of this function of Bcd. It seems significantly less likely that the Cad(Z->B) enhancer would somehow recover activity without recovering chromatin accessibility within at least the anterior ~1/3 of the embryo. We have now qualified this conclusion in the manuscript text to reflect the key assumptions we have made here. We hope that this is a suitable compromise, given the strength of our results with the transgenic Cad (and HC45) reporters and the technical limitations described above.

13) Discussion of model in Figure 7. The idea that in regions where morphogen levels are low, Zld is required to boost activity has been proposed for both Dl and Bcd (Nien et al., 2011). Likewise, that morphogens at high levels can access and activate their high-level targets mostly on their own has also been discussed (same paper). This study corroborates and advances those ideas very nicely, and this should be mentioned.

This has been added to the Discussion.

14) The authors should move the last sentence of the paper “This may be a common property of developmental transcription factors that must gain early access to their target enhancers while the chromatin state of the genome is being remodeled during large scale transitions in the gene regulatory landscape.” to the Abstract.

We have made this change in the manuscript.

[Editors' note: further revisions were requested prior to acceptance, as described below.]

The manuscript has been improved but there are some remaining issues that need to be addressed before acceptance, as outlined below:The authors have satisfactorily addressed reviewer concerns. The efforts made to address further experiments (especially those suggested for the cad reporter) were greatly appreciated.Answers to author questions are answered below:1) Regarding the Kni stainings: We prefer to use the Kni stainings from the first experiment in Figure 1, because that experiment also utilized the same patterning markers (e. g., Btd) used in the other genotypes of the figure. We could move the Supplementary experiment to the main Figure 1, but would prefer to keep it in the Supplement. We have added sentences to the text and to the figure legend to point the reader to the more supplementary material.Reviewer's answer: Yes, please keep it in the supplement.

We have kept this figure in the supplement.

2) Regarding the repressor binding: However, we agree that the data are not entirely clear, and difficult to interpret. Overall we agree that this analysis does not advance the concepts in the manuscript. We did not make it a main figure in the paper for this reason, and one possibility would be to remove the figure and corresponding argument entirely from the last paragraph of the subsection “Bcd binding to genomic targets is concentration dependent”. We would appreciate advice from the reviewers on this point.Reviewer's answer: Please delete the corresponding argument from the last paragraph of the subsection “Bcd binding to genomic targets is concentration dependent”, rather than keeping the new paragraph because it isn't much better with the single repressor focus.

We agree that the discussion of the BDTNP repressor binding is not a valuable addition to the manuscript. We have deleted the text, and panel A from Figure 2—figure supplement 1 has also been deleted.

3) Regarding the assays on cad and HC45 reporters: The major limitation to measuring chromatin accessibility or performing ChIP on reporter constructs (as suggested in experiments 1 and 2 for our cad reporters) is that the measurements must be able to distinguish between the transgenic (reporter) and endogenous (wild-type) copies of the regulatory element in question.Reviewer's answer: While it is possible to distinguish transgenic from endogenous enhancers by having one primer in the reporter sequence (GAL4), which is likely within ChIPed fragments, and one inside the enhancer (as in Xu et al., 2014 for Bcd ChIP on HC45), it is not necessary pursuing this venue at this point.With regard to the new HC45 data, the discussion about endogenous HC45 being accessible whether active or not and with or without Zld or Bcd is interesting, though it is formerly possible that local chromatin at the transgene is not behaving the same way as endogenous. Reviewers recommend having this data as a supplemental figure.

We agree with this point. We will keep the HC45 data in Figure 6—figure supplement 1.